# Human-Caused High Direct Mortality in Birds: Unsustainable Trends and Ameliorative Actions

**DOI:** 10.3390/ani15010073

**Published:** 2024-12-31

**Authors:** Gisela Kaplan

**Affiliations:** School of Science & Technology, University of New England, Armidale, NSW 2351, Australia; gkaplan@une.edu.au

**Keywords:** biodiversity, avian mortality, cat attacks, window collisions, ecosystem services, wind turbines, geothermal and solar energy, contaminations, human attitudes, legal reforms

## Abstract

In the 21st century, more people than ever have a positive relationship with an individual bird or a family of birds, be this as a companion or a regular backyard visitor. People are willing to support birds in the wild, rehabilitate injured birds, and fight against the destruction of their habitat. However, our relationship with birds is double-edged. Each year billions of birds are killed. These are the cases of direct mortality caused via human actions and technology. Direct mortality plays a different and even more surprising role in the decline of avian species than does habitat loss or climate change. Direct mortality indiscriminately affects young and old, inexperienced and resourceful birds alike. This paper first confirms why we need birds in our ecosystems, and then it outlines all major causes of direct mortality. Finally, this paper evaluates how such unsustainable death rates can be reduced by altering the circumstances that cause high death rates in the first place. This may need adjustments to existing technologies or products, and even new legal frameworks. This paper presents a series of targeted strategies that have been or could be implemented nationally and internationally to avoid much of the human-caused direct mortality in birds.


**Introduction**


In 2008, Corey Bradshaw and colleagues identified a significant biodiversity crisis in the tropics [1]. They pointed out that over two-thirds of all known species reside in the tropical (and sub-tropical) belt, calling the decline of species in this belt “a biodiversity tragedy in progress”. The warnings could not have been clearer, and the frustrations of the authors could not have been expressed more overtly. In the same year, Cabrera et al. (2008) [2] tried to identify the world’s most pressing problems via surveys. Most of the replies were anthropocentric; environmental damage did not feature highly in the responses, but at least it was mentioned. Sixteen years after this survey, we can still talk about unabated rates of forest and general habitat loss, rising levels of contaminants and greenhouse gases, and expanding human populations and consumption, i.e., all factors known to be detrimental not only to birds but also to the well-being of humans.

While data consistently report substantial declines in bird populations all over the world [3,4], it is also clear that there are blind spots regarding the many anthropogenic dangers that birds face. Global bird decline may be “just” one of the many symptoms of a changing world and a general decline in biodiversity. Policymakers throughout the world have usually responded to avian death rates as “low priority” compared to “more pressing issues” [2,5], but if they did or still do, this would be an incorrect assumption.

Biodiversity, as must be remembered, has an interactional dynamic. Darwin wrote the following in his *Origin of Species* (1859):

“It is interesting to contemplate a tangled bank, clothed with many plants of many kinds, with birds singing on the bushes, with various insects flitting about, and with worms crawling through the damp earth, and to reflect that these elaborately constructed forms, so different from each other, and dependent upon each other in so complex a manner, have all been produced by laws acting around us” [6].

In a very insightful report of the US National Research Council in 2011, in commemoration of Charles Darwin, the introduction reminded us that biodiversity has two parts: the tree of life, and the web of life, the first dealing with the evolution of all living things (plants and animals), and the second dealing with the interconnectedness of all living things [7]. The lines cited here were true then and are even more relevant now. Donoghue said the following in 2011:

“Darwin and Hutchinson were responsible for solidifying the two great metaphors that orient our understanding of biology—the notion of the tree of life and the notion of the web of life. Each, in his own way, managed quite comfortably, even seamlessly, to interdigitate the two. Today, my sense is that this, unfortunately, is a rather rare ability, and sadly the two have become quite separated from one another in the ways that we teach and carry out biological research. In my view, one of the greatest challenges before us, as scientists, is to work out the ways in which to connect the tree of life with the web of life―to truly reintegrate ecology and evolutionary biology” [8].

Cardinale et al. (2012) [9] argued what was in effect a very similar point, although biodiversity seemed to be perceived here to be in the “service” of humans. The authors agreed that biodiversity underpins the function of ecosystems and then argued that it underpins “the delivery of goods and services upon which humanity relies”. At the least, a dynamic interconnectedness between human existence and biodiversity was established (see also Knight 2010 [10]).

Despite the occasional references and links made to both the tree of life/evolutionary biology and the web of life/interconnectedness, Donoghue’s words, cited above, are important in more than one way. Consideration of the web of life often fades from view despite its central importance [11]. Even “biodiversity” often seems to be cleansed of vertebrates and invertebrates to have become indeed an abstract term.

This paper will synthesise some of the best-documented direct causes of avian death rates and then proceed to query why even preventable and unsustainable death rates of birds are not dealt with consistently and globally (suggested at length below).

The aim of this paper is to show (a) the enormous gulf between our knowledge of the dramatic decline of birds and any concerted national actions that have so far been undertaken for their survival, at least as far as direct mortality is concerned; (b) that many anthropogenic causes of death of birds can be minimised or even eliminated without any major cost to society; (c) why direct mortality is of particular and special biological interest and concern; (d) emerging problems or relatively hidden problems and new challenges with which, predictably, birds are unlikely equipped to deal; and (e) how some of these problems can be alleviated. Ultimately, it is a matter of how much importance we attribute to the living world. Defaunation has increased in all vertebrate species and across land and oceans [12], but possibly with the highest losses in avian species [13,14], as Figure 1 below suggests.

This paper will also relay findings that explain the role of birds within any ecosystem, usually referred to as ecosystem services [15], and how and why continued further extinctions on a global scale will also be detrimental, if not disastrous, for human populations. Such insights are now more commonly expressed, and this represents one way of identifying why we need birds [16].

Importantly, this paper includes publications that have made the case for birds’ survival, regardless of discipline. In reading beyond science publications, this paper proposes that other professional groups—especially those working in architecture, technology for alternative energy, communication, and the law—have a very important role to play in changing the fate of wildlife (here specifically, birds).

## 1. Section I—The Human–Birds Context

### 1.1. How Humans Perceive Birds

There are many publications on birds as companions. It seems that pet ownership of birds may also have lasting benefits, similar to those that have been recorded for cat and dog ownership [17]. Many writers have shown that the beauty of a peacock, the flight of an eagle, or the song of a nightingale can be something to which humans relate [18]. Indeed, many have enjoyed birds for aesthetic reasons [19,20], be this because of their beautiful or extraordinary plumage [21], for their exceptional song (either their musical song quality and/or repertoire size [22]), for status reasons [23], or for sheer adventure value. Just watching birds in one’s own backyard can be pleasing [24]. Another way to “enjoy” birds is to watch them in their natural habitat [25]. Such activities have been shown to deliver health benefits for humans [26,27]. In many books, including well-researched books such as *Birds and Us* [28] that cover major historical periods, it is argued that birds have enhanced our lives in a myriad of ways [29].

However, it has also been recognised more recently that pet bird–human relationships may differ from those in cat or dog pet ownership. Recently, researchers have developed a bird-specific scale to be able to quantify aspects of such human–bird relationships [30]. One reason for this difference is that birds, even pet birds, are not usually classified as “domestic” animals, except for poultry and pigeons. This has to do with the ancient history of poultry and pigeons (involuntarily) entering the human world for food (chicken, ducks, geese) or hobbies (pigeons), often thousands of years ago [31,32,33].

Most pet birds, however, have had a much more recent history with humans. They may have been taken directly from the wild or have had only one or two generations in captivity of any kind (in private households, zoos, or research facilities). Even the most loved pet birds may thus still have behavioural markers of their wild counterparts and not be fully adapted to captivity, and some may continue to suffer as a consequence of having been removed from the wild [34]. Recent introductions of wild birds to pet status, however, have so far not been undermined by changing their appearance and general traits by creating new breeds with different colour and plumage patterns, as is the case in domestic chickens and domestic pigeons. Domestic pigeons (*Columba livia domestica*) now have at least a derived 100 breeds, while the wild rock pigeon (*Columba livia*) still occurs widely across the globe [35,36]. The domestic chicken (*Gallus gallus domesticus*) has resulted in even more breeds than the pigeon, with its wild ancestor, the smaller red junglefowl (*Gallus gallus*), still occurring in pockets of South and Southeast Asia and on various islands [37].

Despite evidence of widespread positive human relationships with birds [38], the reality is often far less heart-warming for avian species in their natural environment. We now know that their survival into the future is at stake. Indeed, the level and speed of decline in bird numbers are distressing and alarming [39,40,41,42], and data show clearly that, as a group of animals, birds in particular are predicted to continue to decline rapidly and in large numbers (Figure 1). The *Living Planet Report* of 2024 [4] showed a catastrophic 73% decline in the average size of monitored wildlife populations over just 50 years (1970–2020), with the steepest declines in Latin America and the Caribbean (−95%), Africa (−76%), and the Asia–Pacific region (−60%) [4]; alas, all of this has to do with human behaviour and actions.

Finn et al., in 2023 [14], wondered whether an exclusive focus on the IUCN conservation category of “currently threatened”, without consideration of dynamic population trends, may underestimate the true extent of the processes of ongoing extinctions across nature. They therefore took the number of species decreasing in the “near threatened” category, based on data derived from the IUCN Red List. Note that these figures (Figure 1) are estimates [3] and concern only monitored species, i.e., not all taxonomic groups received the same level of evaluation; hence, the Red List should not be interpreted as a full and complete assessment of the world’s biodiversity (see also [4]).

#### 1.1.1. Aloof or Negative Attitudes

In many ways, cities have been hostile places for wildlife, including birds. Nearly half of the world’s population lives in cities now, and these urban spaces are expanding at twice their population growth rate and projected to increase further [43]. Even the megacities are still predicted to grow in the future, and so is the number of people living in them [44]. But increasingly, it has been found that many aspects of city structures and life are unsustainable [45]. Cities and suburbs are concreted in (also called “concretosis”), i.e., most urban and even suburban land consists of sealed surfaces for housing, roads, and industry [46]. Parking lots make up a surprisingly large number of vacant spaces and, incidentally, many sealcoats of such parking spaces contain polycyclic aromatic hydrocarbons (PAHs), which are known to be acutely toxic (carcinogenic) to humans [47] and likely also to animals, including birds.

We also know that such conglomerate steel and concrete spaces are heat traps, especially when they have few or no trees and other green spaces [48,49,50]. Despite this knowledge, there still exist landscape gardening/architecture companies that market outdoor spaces of properties to be designed in concrete and pebbles, adorned with cacti and succulents, preferably spiky and lacking in any fresh water, often including tiny patches of artificial grass as ground cover. No single native bird—and not even worms, insects, or butterflies—could benefit from such an environment. It is purposely hostile to any living organisms and to native animals but is marketed as “low maintenance” (see any website on concrete landscaping), and as research has also established, the lower the number of green spaces in a city, the lower the bird count will be [51].

Many of these cities (and megacities) have thus become human fortresses, not in the sense of crime control and gated communities (although that too [52], but here in the sense of being shut off from nature, minimising encounters with nature and any knowledge about nature [53], with some drastic results. Surveys assessing basic knowledge of animals among high school students revealed they did not all know that milk was produced by cows [54]. Adolescents in south–central Los Angeles, as Miller (2006) commented, were “more likely to identify correctly an automatic weapon by its report than they are a bird by its call” [53].

The birds that stay in the city against all odds are not always welcome. A rare European urban study showed what types of birds people would tolerate [55]: tits and woodpeckers were acceptable, but crows or owls were no more acceptable than mice [55]. Most of these dislikes are based not on knowledge but on some kind of misinformation or even mythology. In particular, crows and ravens are still subject to the myth that they bring bad luck [56], and such myths have even been passed on to younger generations. Some studies have also shown that associations with humans in urban environments can end fatally for birds. Barrett and colleagues [57], for instance, found a relationship between avian cognitive abilities and likely hostile human reception. To quote,

“Animals that are the most adept at acquiring anthropogenic resources, and those that exhibit high levels of cognitive abilities such as boldness, learning, innovation and behavioural flexibility, may also be the most at risk for lethal encounters with humans” [57].

A survey in New South Wales, conducted in 2002 by the Department of Environment and Conservation [58], asked people about their attitudes to native wildlife in their urban space to establish how proactive the message in favour of a greening of urban spaces would need to be. They found that because people generally hated spiders, snakes, bats, and even bees, they did not want a garden that could attract any of these unwanted species. Apart from very small songbirds, the majority wanted no other, larger birds in their private space [58]. An American survey found complaints about specific species, such as the house sparrow (*Passer domesticus*), European starling (*Sturnus vulgaris*), and blue jay (*Cyanocitta cristata*), that attracted people’s attention for their allegedly negative qualities, such as their vocal behaviour and effects on personal property (e.g., bird defecation [59] and such criteria put birds in the ‘Not encouraged’ group (Table 1).

A Dutch study showed a great popularity of paved courtyards [61] in a population where over 90% live in cities. In 2007, in a US study, Clayton showed that gardens are not perceived as part of a larger ecosystem. Convenience (ease of maintenance) was a priority [62].

Importantly, as some surveys had shown a lack of interest in native wildlife or birds in particular, Fančovičová and colleagues [48] were right in suggesting that (a) a change in attitudes to wildlife must begin with education at the school level, and that (b) before even beginning to talk about wildlife sanctuaries in cities, there need to be outdoor programmes teaching about native plants, how to plant them, and what they do. Trees are not enough and will not remain on their own for long. People’s attitudes to wildlife also partially determine the plants to be (re-)introduced. Such programmes have been conducted widely across the world because it has been recognised that these concrete environments are bad for human health, and some might even believe that it is important to retain, regain, or reintroduce biodiversity.

Dunn and colleagues [63] concluded, even if seemingly with a tinge of regret, that

“Although most ecosystems and species will not be saved in cities, their conservation may depend on the votes, donations, and future environmental leadership of people in cities; so, in the end, a great deal depends on urban nature. The urban jungle, with its many non-native species, may well be the breeding ground for future environmental action.” [63].

#### 1.1.2. Positive Attitudes and Actions

Even the densest city can potentially house five principal habitat types (municipal greenery, private gardens, balcony plants, forest remnants in small parks, and planted edges at brooks, creeks, or rivers running through cities). In the past two decades, developments towards “green cities” have been promoted widely across the world, but when assessing the parameters used for the term ”green” city, they are largely for construction methods, technology, and reducing energy consumption by clever designs or alternative tools (from petrol to electric), although biodiversity is now also embedded in debates on ”greening” [64].

Do people, especially in cities, care about birds at all? And would or do they care if they learned that birds might die because of human actions? The answer is clearly ”yes”. In the intervening years of a decade or more of changed school education curricula and public statements about the environment, there has been an increased awareness of and literacy in environmental and ecological issues amongst the general public, based on changing social values and attitudes towards nature [65].

A substantial number of researchers, activists, and a surprisingly large number of members of the public are now devoted to the welfare and survival of native animals, with a demonstrable interest in native wildlife. Interest in birds is particularly strong, as evidenced in countless special species study groups, as well as local and national bird organisations, be this for the scientific pursuit of specific species groups, birdwatching, citizen science engagements or bushwalking with a focus on birds, or even bird conservation. In the USA alone, according to the 2016 US Fish and Wildlife Services Report [66], over 45 million Americans apparently take part in birdwatching, and they spend approximately USD 41 billion on related trips and equipment, contributing significantly to local communities and the national economy as a whole. Interest in birdwatching now ranks globally before gardening and fishing in many countries. The UK is home to the largest bird organisation in Europe, the Royal Society for the Protection of Birds (RSPB), with more than 1 million members, and that society claims that around 6 million United Kingdom residents are regularly engaged in birdwatching [67]. Most countries have developed specialist tourist niches for birdwatching, be this for the “hardcore” enthusiast or occasional casual birders with very special interests.

The number of people volunteering, working, and writing in the field of conservation is equally high in Australia [68], showing that about 8.1 million Australians are involved in some form of volunteer work for nature (at a population of 24 million) and at least 20% of them belong to wildlife organisations. Then there are the professional and academic conservationists, who may be guided by very different ethical principles and may thus vary enormously in approach, but they all have an expressed interest in native wildlife, including birds [69,70,71,72,73].

Many volunteer groups can be loosely grouped under environmental stewardship. The term environmental stewardship has been used to refer to such diverse actions as creating protected areas, replanting trees, limiting harvests, reducing harmful activities or pollution, creating community gardens, restoring degraded areas, or purchasing more sustainable products [74]. There are three types of environmental stewards: doers, donors, and practitioners. Doers go out and help the cause by taking action. For example, the doers in an oil spill would be the volunteers who go along the beach and help clean up the oil and rescue impacted animals [74]. In all, though, many of these groups committed to faunal biodiversity tend to be situated somewhere in the human social network that is nowhere near positions of political influence [75]. Despite this substantial and declared interest in birds by millions of people, the world of birds is not intact at all, and species are sliding into the dangerous territory of extinction risk. The passion, compassion, and general positive attitude to birds is not enough to save them. The question is whether we need them at all, and who should be asked to make changes in favour of avian survival if we do.

### 1.2. Why We Need Birds

Ironically, the best evidence as to why we need birds arises in their absence. When avian species populations decline, as research has found, in extreme cases such decline or absence can result in a breakdown of ecosystem processes and services [76,77], and it can even have implications for human societies [9,10]. We could see catastrophic insect plagues that can overwhelm the land and lead to harvest failures, catastrophic die-backs of forests and grasslands, and the spread of insect-borne diseases, to name a few problems.

How would one measure whether and how birds are useful (and perhaps even important)? This took surprisingly long to devise and finally resulted in assessments of avian “value” according to so-called “ecosystem services”. These were included and first standardised in the United Nations Millennium Ecosystem Assessment (2003). This was in some ways disappointing, but at least it formalised categories that could be measured [78]. The basic (four) principle ecosystem service categories are anthropocentric and betray a deep-seated belief that most such “services” are provided by or are of direct benefit to humans: 1. provisioning services, e.g., food production; 2. regulating services, such as climate and human disease; 3. cultural services, such as spiritual enrichment and recreation; and 4. ”Supporting services”, including what happens in nature, such as ecosystem processes.

The term ecosystem services has persisted even though it is slowly imploding from within [79]. The idea of “services” implies that the world exists for humans, and that some ”services” are more useful than others and we are free to decide on the relative merit of who should be kept alive. Given that humans were one of the latest additions to life on Earth, I would prefer to think of “roles” rather than “services” of wildlife/birds, and once one changes that one word, one can meaningfully speak of biodiversity and the natural environment. Gaston (2022) [80] seemed to express a similar sentiment when he wrote the following 20 years after the UN assessment scale:

“What do birds do for us? Some may find this an inappropriate or perhaps distasteful question, suggesting as it might that the importance or value of birds lies principally in the ways that they benefit people, and with perhaps an unspoken implication that if they do not do enough then we should not be too concerned as to what befalls them.” [80]

Indeed, these underlying beliefs are crucial to unmask. David Suzuki [81] warned one-quarter of a century ago that we needed to adapt to a different way of thinking, and quickly, to avoid disintegration. He wrote the following in his book *The Sacred Balance*:

“we have shifted from the notion of embeddedness and responsibility to a belief that our great intelligence makes us the most important animal, endowed with the ability to extricate ourselves from the web of relationships to a position of exploitation, control and management” [81].

As early as 1909, a concerned American ornithologist, F.J. Wenninger, writing about some declines in birds, tried to indicate that birds are indeed important in our “web of relationships” as Suzuki put it, but he did so in a good positivistic fashion by summarising the “economic” value of birds [82]. He brought together examples showing their usefulness, such as the capacity of birds to “destroy” insects, pointing out that a potential insect load can destroy a tree if left unchecked. For instance, apple trees had a potential insect load of 176 species, oak trees 537, willows 386, birches 297, poplars 264, beeches 154, and pines 100 [82]. Of course, some insects are more damaging for trees than others, and even a single species can overwhelm a tree or shrub. Wenninger (1909) found that cuckoos consume 50–400 caterpillars per day, chickadees eat 200–500 insects or 4000 worm eggs per day, and the meadowlark (*Sturnella magna*) feeds on 72% of insects, chief of which are the very harmful grasshoppers, locusts, and crickets, as well as beetles and bugs. He further quoted five insect-eating birds in a Massachusetts study that found that their daily consumption totalled 2.5 billion obnoxious insects per day [82]. But many of these natural foraging behaviours in birds have been undermined and turned against birds, because agriculture has stepped in with lethal pesticides that kill the very birds that would help reduce the numbers of damaging pests [83,84] (see more on toxicity and poisons below).

That aside, birds (and bats) take care of insect surplus and potential pests. Insect-feeding birds also save vast grasslands, such as the Australian magpie (*Gymnorhina tibicen*) feeding on scarab beetle larvae, an invasive species that changes grazing lands into wastelands [85,86]. Forests are at constant risk of decimation by insect pests. Birds fulfil a vital role here too, sometimes concentrating on small insect pests. Larger birds may also tackle potentially defoliating pests, such as some large stick insect species (Phasmatidae), which currawongs (*Strepera graculina*) can keep under control or remove almost entirely from such a forest [87]. Even species such as the aggressive and highly toxic bulldog ants, *Myrmecia tarsata*, are eaten by currawongs. The much maligned sulphur-crested cockatoo (*Cacatua galerita*), corellas (*Licmetis/subgenus of white cockatoos)*, and galahs (*Eolophus roseicapilla*)—all species internationally noted for their exceptional cognitive and problem-solving abilities, as well as social and parental skills [88,89,90]—are not widely known or acknowledged for their usefulness. Little, if anything, is ever said about sulphur-crested cockatoos (*Cacatua galerita*) ripping flaky bark off native eucalypts and exposing tree-destroying grubs, which they consume. The corellas, of which the little corella (*Cacatua sanguinea)* may be the most numerous, as well as the sulphur-crested cockatoos and galahs, feed on onion grass corms *(Romulea rosea* L.) and remove this invasive weed from grasslands and agricultural fields [91,92].

Should these feeding habits of birds be called an “ecosystem service”? Perhaps, but feeding is a behaviour that evolved in many bird species as a response to readily available and diverse food sources, or in response to a scarcity of familiar food items, which required the discovery of new food sources to survive. Equally, should the role of birds in seed dispersal or pollination be singled out as a “service”, and to whom? Plants depending on pollination have evolved to produce attractants that ensure that they are visited, and some of these plant methods are quite sophisticated, e.g., producing just enough nectar to satisfy a bird or insect for a short time but then requiring a bird’s or insect’s return, or by producing flowers in a shape that can only be accessed by a limited number of species, as extensive research into nectar-feeding birds has shown [93,94]. Some 900 avian species pollinate plants [82], and the birds, in return, receive valuable and nutritious food. These are co-adaptations resulting in symbiotic relationships in which the flowers play an active role by developing curvatures and lengths of buds suitable for just a small range of species—sometimes only one species, as for instance in the Andean sword-billed hummingbird (*Ensifera ensifera*), drinking nectar from flowers with long corollas. This avian species co-evolved with the plant species *Passiflora mixta* [95]. Indeed, as research on hummingbirds has found, the evolution of bill morphology is likely to have been driven by a small subset of the flowers visited by very specific hummingbird species [96]. Equally, seed dispersal is accomplished by the bird consuming the flesh of a fruit. Currawongs, while switching food regularly, are largely frugivorous [97]. They consume nutritious berries, but their gut is organised in such a way that the stones and pips do not go through the entire digestive system but are collected in olive-stone-sized and -shaped pips that are regurgitated and eliminated via the beak. The seeds in such a “package” have their own composted, moist supply of nutrients, are often transported far away from the parent tree or shrub and are thus given the best chance to grow [98]. One can give hundreds of such examples that demonstrate the essential life-support functions provided by birds (e.g., seed dispersal, pollination, removal of insect pests and weeds, aerating or fertilising soils, saving trees by removing borers and other insect pests). Knowledge of such behaviours confirms the dynamic interrelationships between soil, plants, and birds, and these are sound biological parameters. They also include pest control or pest regulation, as has been shown repeatedly [99] and has even been calculated as potential replacement costs. For instance, Eurasian jays (*Garrulus glandarius*) have been credited with the survival, expansion, and growth of one of the oldest stands of giant oaks in Europe (*Quercus robur* and *Q. petrea*), in the Stockholm National Urban Park [100]. Hougner and colleagues (2006) actually calculated the value of one pair of Eurasian jays down to dollars and cents [100]. For the ecologist and ethologist, the example of the European jay is but another example of the importance of mutual relationships between mobile link organisms (e.g., birds) and keystone species, and it is this dynamic that generates high biodiversity benefits, not just with trees but also in symbiotic and mutualistic networks with grasses, fungi [101], and even ants [102]. Mutualistic arrangements between birds and plants and/or micro- organisms have also been shown to contribute to the regeneration of forest and non-forest urban patches [103].

Despite these well-known facts of co-evolution and mutualistic arrangements, the ecosystem services model may have survived for so long because it is useful for policymakers, business leaders, farmers, and chiefly politicians, who may need some form of rationale for why bird conservation in land use and even in urban development is so important. Since Darwin may not have been on their reading list, and the dynamic link between evolution and ecosystems may not be apparent (and, hence, not considered important or known), the natural ecological processes may indeed need “translation” for political purposes. As Wenny and colleagues (2011) rightly argued a decade ago, far too many public figures and politicians might still misinterpret conservation efforts as a “luxury” [104].

Judging by the paucity of public funding generally allocated to conservation efforts, one can see that ongoing arguments are needed to bring wildlife and forests back to the frontline of political debate. In 2021, the International Monetary Fund commissioned an independent report, called “Building Back Better: How Big Are Green Spending Multipliers?” [105]. The report found that “the estimated multipliers associated with green spending are about 2 to 7 times larger than those associated with non-eco-friendly expenditure, depending on sectors, technologies and horizons” [105]. While this may sound impressive, and even convincing, these figures include all key carbon-neutral or carbon-sink activities—from zero-emission power plants to all of the new technology utilized for the reduction of carbon emissions. Nobody doubts the urgency of these endeavours. However, if the wildlife and birds are short-changed, as they are by and large at the moment, the technology by itself will not save the planet, despite the seemingly unshakeable faith of humans in our own technology and innovative abilities [106,107,108,109]

## 2. Section II—Anthropogenic and Related Causes of Direct Mortality

Major causes of death in birds are extremely well documented and will be described below in some detail. This section is subdivided into seven major causes (cat kills, window collisions, impact of cars, alternative energy (wind turbines; solar; c—geothermal), power lines, toxicity and pollution, and finally illegal hunting and poaching). Such an outline is necessary here to indicate the breadth and depth of compounding aspects of risk of death for birds. To alleviate any of these problems and improve the survival of birds, we need to know, be aware of, and fully grasp the extent to which birds are threatened every single day by having to contend with countless anthropogenic dangers, in addition to a range of natural predators with which birds might have co-evolved. Section III of this paper will then offer an interpretation of these data and show that there are possible and even quick solutions to reduce the death toll for some of these problems.

It has been shown that birds in particular have experienced and are still experiencing precipitous population declines across the globe as a result of multiple anthropogenic stressors [3,4,110]. In the United States, 100 bird species and subspecies are listed as federally threatened or endangered [111]. Hence, the problem is not just one of overall numbers but of species declines into the precipitous territory of potential extinction [112].

According to USA records, the top three most serious and impactful causes of bird fatalities are cat attacks, collision with windows and, collision with cars, in that order. These findings seem to be mirrored in other countries, as in Australia (see Figure 2).

Humans have transformed habitats for their own purposes, and even semi-rural and rural areas are now criss-crossed with power structures, roads, and cars. Also, cats, be they unowned (feral) or pet cats, exist alongside birds in these same areas. Moreover, the records of death are not isolated events but are cumulative; thus, such bird declines have effects on the ecosystem [110].

### 2.1. Kills by Pet Cats (Felis Catus)

First and highest on the death list are pet cats [114,115]. Statistics suggest that a staggering number of deaths occur every single year and, if anything, the number of deaths of birds is rising due to rising human populations, higher density of buildings, and increased cat ownership. For the USA, the death rate of birds per year has now reached 1.5 billion, equivalent to one-fifth of the entire global human population (Figure 2).

To arrive at a relatively reliable total number of bird kills per cat, even in urban environments, one usually relies on surveys of cat-owning households. A study in Bristol, England, used a small area and established the cat density in that area (>200 cats/km^2^), and then used household surveys in that plot and relied on feedback from owners that their cat had actually brought home the prey item [116]. Cat owners, according to a survey by the Australian Companion Animal Council (ACAC) [117], admitted that their well-fed cats had killed an average rate of 3.3 animals per month, or 40 animals per year, and these were only those cases when a cat displayed the dead birds to the owners. Multiplied by the 2.5 million domestic cats known to live in Australian households, even this conservative estimate adds up to staggering numbers of prey killed per year.

As for non-urban environments, one early study of cat-caused deaths in birds by Rose (1975) investigated cat kills in a town near a national park (Wahroonga near the Ku-rin-gai Chase National Park, New South Wales), showing that domestic cats predated on native birds not just at high rates but on many different species [118]. Some 35 different species were taken, and these were not just small birds but also medium-sized and larger species such as the crimson and eastern rosellas (*Platycercus elegans* and *P. eximius*) and red wattlebirds (*Anthochaera carunculata*). Mammals taken even included ring-tailed possums (*Pseudocherius Peregrinus*), which may weigh up to a kilogram and should thus be substantial opponents for a cat. Evidence of pet cat invasions into protected national parks is particularly worrying and, as far as one can tell, no large-scale actions of any consequence have been undertaken to reduce the carnage. A major review, prepared for the Australian Nature Conservation Agency in 1996, provides details of the impact of feral cats [118], but to solve the problem of their invasive role in national parks and elsewhere is indeed a separate and quite complex issue, beyond the scope of this paper. However, it is important to note that unowned/feral cats add to the death tally for native birds.

The most detailed Australian figures to date on domestic cats and their hunting behaviour in Australia were published in 2024 by the Biodiversity Council of Australia (BDC) (see Table 2 below), suggesting that Australia now has 5.3 million cats, doubling the number of cats identified just 20 years ago, and also showing that every third household in Australia is now registered as a cat owner (BDC 2024) [119].

**Table 2 animals-15-00073-t002:** Fast facts on pet cats in Australia in 2024 (Biodiversity data: [120]).

Total pet cat population	5.3 million
Percentage of households with pet cats	33%
Size of animals cats can kill	up to 4 kg
Percentage of pet cats that roam	78%
Percentage of roaming cats that hunt	71%
Mean home range of a pet cat	2 hectares
Average density of roaming/hunting pet cats in Australian suburbs	54 to 100 per km^2^
Average number of animals (mammals, birds, reptiles) killed per roaming cat per year	186 million
Number of all mammals, birds, and reptiles killed by pet cats per year	546 million
Number of native mammals, birds, and reptiles killed by pet cats per year:	323 million

The number of bird deaths as a result of cat attacks, however substantial it appears, may yet be a significant underestimation. In my experience, pet owners have tended to be reluctant to believe that their well-cared-for pet may do any harm to local wildlife when it roams the streets or the undergrowth. One characteristic of cats is that they do not necessarily kill or eat their prey immediately or take it home, but they will “play” with their prey and, in so doing, may injure or kill it [120].

Many birds may even get away alive from their feline attacker. In those cases, there may be no visible signs of injuries post-attack. Owners may not bear witness to the attack or, if they do and see a bird flying away post-attack, may falsely conclude that their cat has done no harm. They cannot see the deadly results because the victims (largely birds) do not necessarily die immediately as a consequence of such an attack. It is enough to kill if the cat’s claws or teeth scratch and penetrate the skin of a bird, because cats carry bacteria that are toxic to birds and can even be serious for humans, both on their claws and in their mouths, e.g., *Pasteurella multocida* and, less often, *P. septica* [121]. A cat attack, unless the infection is treated immediately with antibiotics, invariably ends in the death of the bird, but even such minor and usually invisible scratches take days to take effect [122]. Death in birds occurs usually between 3 and 5 day after the cat attack (personal records), and it is not a quick or pain-free death [123,124]. *Pasteurella multocida* invasions cause a range of symptoms, including septicaemia, diarrhoea, and neurological signs, including slanting neck, head tremors, movement disorders, feather ruffling, anorexia, oral mucus discharge, and a visibly increased respiratory rate [125]. When death occurs days later, it tends to happen far away from the culprit that caused the demise [126]. One can therefore easily dissociate the cat attack from the death of the bird, which may be dying somewhere alone, usually hiding on the ground.

Very few studies make clear the often invisible relationship between cat attacks and the death rate of birds. For this reason alone, it is suggested here that the number of actual deaths per annum might well be much higher than even these staggeringly high estimates suggest.

### 2.2. Window Collisions

In terms of mass deaths, window collisions are second-highest on the list of causes of avian casualties [127] (see also Figure 2 above). Window collisions have been increasing over the years. This has to do with the design and height of buildings. Windows have also progressively become larger. Indeed, many high-rise buildings now sparkle in steel and glass. There are now higher percentages of glass in façades and more continuous surfaces of glass than at any time in the past [128,129,130,131].

Windows, as a product, have become very sophisticated, such as creating heat barriers and offering new alternative designs for eco-efficient solutions [132], but not yet with respect to inbuilt characteristics designed to warn birds and make them veer away in time before crashing into a window. Indeed, trends in modern architecture have gone in the direction of being more harmful to birds than ever, at least in terms of the increased size, height, and reflectiveness of windows. The “Modern” style has meant far too often that high-rise office blocks in cities tend to consist of aluminium and glass [133]. Fashionable, architecturally designed private homes have also increasingly exchanged walls for glass, at forbidding cost, but allowing views of nature into the building. Private homes, especially at the high end of the market, use tons of specially glazed glass, be it for their structure, walls, exceptionally large windows, or even ceilings that allow light to flood the rooms [134,135,136], and it seems that these modern glass structures and materials are particularly “guilty” of causing bird–window collisions.

### 2.3. Car Collisions

Birds are regularly hit by cars, but many that are foraging beside the road also die as a result of the strong wind impact of a car at speed, which causes them to tumble backwards, causing fractures or even death. Large eagles attending to roadkill usually respond quickly and try to get away when they see an oncoming car, but because of their size, their take-off is often too slow to clear the approaching car. Fewer birds are killed by cars than by cats, but as Figure 2 above shows, the death rate from car impacts is still countable in millions of birds lost. The figure of 300 million per year in the USA is certainly a substantial number [115,137], and other countries have recognised the loss of birds as significant enough to start their own investigations [138,139,140,141,142]. Roads in New South Wales alone, just one state in Australia, claim 7000 native animals as victims daily. One of the few studies on road kills in New South Wales was conducted by WIRES (Wildlife Information and Rescue Services) in conjunction with Professor Cooper of Macquarie University, showing that the animals killed on roads consisted largely of native animals (80 different species in a sample size of 381), and mostly birds [143].

Although, so far, these three causes of direct mortality of birds are statistically captured as being greatest in number, other direct mortality data add considerably to the toll, and it is yet to be determined whether some of these additional sources causing instant deaths may at some stage be equal to the three above. One disconcerting set of data on the fate of birds concern alternative energy, be it wind or solar, and the associated substantial expansion of powerlines that a full network of alternative energy will require in the coming decades.

### 2.4. Alternative Energy

The hunt for energy sources beyond fossil fuels has led surprising quickly to many innovations utilising the power of the wind and sun in particular [144,145] and opened up new markets and career opportunities. For many people, these can indeed be regarded as praiseworthy alternatives. It seems that wind turbines are here to stay, because they are considered central to a sustainable energy future, phasing out fossil fuels as part of the energy revolution [146]. These new ideas have focused on energy production using water, sun, or wind. Solar panels are now common on the roofs of houses, and in recent decades, solar farms of often substantial size have been installed on all five continents. Likewise, wind farms have been installed everywhere, found across Europe, Asia, and the USA, and are beginning to be widespread in Australia. By choice, these wind turbines are now in their third generation, becoming bigger, the blades moving faster, and with more substantial energy returns. They produced about 0.1 GW per turbine in 2021, and the aim is to reach 360 GW by 2050 [147]. Wind and solar energy farms entail large national investments in the energy industry, controlled by powerful companies, and are signed off on by governments for the long term. These alternative power sources promise to reduce or end the dependence on fossil fuels.

#### 2.4.1. Wind Turbines

Bladed wind turbines, be they on- or offshore [147,148], are increasingly blanketing the countryside and patches of ocean. It would not be an exaggeration to suggest that the presence of these structures and their bladed action may likely rank somewhere among the three most substantial causes of direct mortality in birds in the near future if the current speed of installations continues, and if no further adjustments are made to current designs. As outlined in detail elsewhere [149], since their introduction in the last 25 years or so, blade strikes have killed literally millions of birds and bats around the globe [149].

On an annual basis, hundreds of thousands of birds and bats also suffer terrible injuries inflicted by bladed turbines and, presumably, these victims eventually die [150,151,152]. A well-documented case is Spain. Spain had about 18,000 bladed wind turbines ten years ago, rising to more than 22,000 in 2024. The annual death toll of birds in Spain alone was about 6–8 million birds and bats; that is, approximately 110–330 birds and 200–670 bats per wind turbine were documented to have died each year. English newspapers [153] talked about “the shocking environmental cost of renewable energy”. And at just one Australian wind farm in Tasmania, eagles, falcons, and other raptors accounted for a disproportionate number (almost one-third) of the estimated 1500 birds killed there each year. As a consequence, concern has been expressed that the Tasmanian eagle could go extinct [154]. South Africa has reported similarly devastating bird losses due to wind farms [155]. Offshore wind turbines, and especially those in the path of migratory land and seabirds, especially when placed closely together, also cause serious injuries and fatalities of birds [148,156,157].

In the meantime, both in Spain and elsewhere, alternative models, called wind generators, have been invented. Technically referred to as vortex-induced vibrational, aeroelastic resonant wind generators, the difference is that they harness energy by oscillation. They are bladeless, produce little noise, and have reduced maintenance costs. Vortex Bladeless operates on low-to-medium wind speeds, is considered energy-efficient, and apparently generates about the same amount of energy at a cost 45% lower than that of a conventional three-blade wind turbine [158,159,160,161], but they are as yet rarely used. Australia and many other countries have instead opted for the largest, most destructive bladed model of all available on the market, and that choice has so far more or less guaranteed the ongoing slaughter of birds.

In some cases, the evidence of bird deaths and mutilations has led to the innovation of an apparatus that can be retrofitted to wind farm sites, which is meant to stop the blades from turning when a surveillance camera has spotted wing movements and to automatically switch on again once the birds have passed. At the Altamont Pass Wind Resource Area in the Rocky Mountains (a main migratory route), for instance, mortality rates and mitigation attempts have been monitored for many years, and the evidence has mounted that, indeed, birds are killed in large numbers, especially large raptors [162]. The iconic American golden eagle (*Aquila chrysaetos*) has been among the most documented victims of bladed wind turbines [163,164].

Retrofitting additional warning devices when birds approach the turbines was thought to solve the problem of eagle fatalities and deaths of other species [165], but the success of this method has also been doubted [165]. Other proposals include using deterrents such as light and sirens [166], as well as geofencing [167]. These matters of retrospective mitigations do not necessarily do what they promise. Typically, no compliance with any regulations is required, underreporting may occur, and poor research design—unintentionally or by design—can obscure the true figures [168]. Secondly, not all bats and all species of birds behave in the same manner when nearing collision with deadly wind blades. Brown and colleagues (2016) [169] argued that the protective devices may have fostered reduced fatality rates for select raptor species but might have inadvertently increased fatalities of other birds [170], partially because the technology only tested responses by raptors.

Even though bladed wind turbines do not rank amongst the three worst causes of bird losses yet, this special case has been described here in some detail because it would suggest a disregard of bird and bat deaths in favour of perceived needs, profit, and perhaps a belief in the inevitability of the solutions that we now have. Publications were already available some 15 years earlier that warned of the potentially disastrous effects on birds and bats [170,171] in terms of loss of life or, as has been established now, also loss of habitat [172]. Moreover, alternative designs without the dangerous blades were already available, and yet bladed wind turbines are becoming larger, taller, and deadlier, seemingly regardless of the known consequences. It seems that this is one technological invention that could be made safer and more environmentally friendly.

#### 2.4.2. Solar Energy

Solar energy has a much longer history, with the first solar panels appearing on individual household roofs dating back to the late 19th century [173], and it was hailed as a democratisation of energy supply [174].

The industrial-sized solar power plants that are now being installed tend to be huge. Roughly 4 to 6 acres is needed per megawatt (MW) of installed capacity. A 5 MW solar farm would thus require approximately 20 to 30 acres (8 to 12 hectares) of land. Installations have happened in deserts and, in the US, many are planned to be placed on farmland [175].

Solar power plants (SPPs) are divided into photovoltaic (PV) and concentrated solar power (CSP). Photovoltaic (PV) technology uses semiconductor cells to convert solar energy into electricity, and cells are assembled on panels that facilitate installation at energy facilities. Concentrated solar power uses reflected sunlight to generate thermal energy. The latter is apparently less common in the US, and development trends have moved away from concentrated solar power (CSP) to PV facilities [176]. Utility-scale solar energy (USSE) capacity is expected to increase six times more than wind energy capacity between 2020 and 2050, and this makes solar energy capture a topic of vast implications for land use, food production, habitat loss, and native animal/bird survival.

As of 2022, there are approximately 5500 major solar projects across the US, with existing installations generating 55 GW, and projects under construction or in development generating 110 GW [177].

These mega-installations raise the question how animals generally and birds in particular cope with miles and miles of solar panels affixed just above the ground. Solar energy can impact avifauna directly by injuring or killing birds that collide with photovoltaic (PV) panels or reflective concentrated solar power (CSP) heliostats, or with associated infrastructure. At concentrated power tower facilities (CPs), birds may also be burned or incinerated when they fly through concentrated beams of solar flux (air temperatures in such an area may reach more than 800 °C). This effect may be exacerbated if the reflective surfaces making up the solar hardware serve to attract birds to the area [178].

The results are so far very inconsistent, however. Studies in the United Kingdom [179] and Germany [180] have argued that species diversity and bird abundance at PV SPPs are higher than on adjacent non-built anthropogenic land, and some have even considered the new solar installations to be “conservation measures” [181].

On the other hand, studies conducted in fully functional solar farms have presented a very different set of results. It has been reported that solar energy can impact avifauna directly by injuring or killing birds that collide with photovoltaic (PV) panels or reflective concentrated solar power (CSP) heliostats, or with associated infrastructure. At concentrated power tower facilities, birds may be burned (or incinerated), as said before, when they fly through concentrated beams of solar flux [182]. At one site (Ivanpah), trauma and solar flux injuries were the leading causes of avian mortality. Exposure to solar flux resulted in singeing of feathers and, depending on the severity and length of exposure, resulted either in immediate death (catastrophic loss of flying ability) or delayed mortality because of flight impairment. Kagan and colleagues (2014) [183] described the site as a “mega-trap” because the warmth attracts insects, the insects attract birds, and the injured, dying, or dead birds attract predators—avian and mammalian, from small to large—of many different species [183]. According to Horváth et al. (2009) [184], the attraction of insects is also due to the polarisation of light above PV SPPs [29] and, thus, is a problem not specific to concentrated power tower facilities [184].

The more commonly used utility-scale solar energy facilities are also a source of direct mortality of birds. Systematic avian mortality information is available for three USSE facilities in California. Annual USSE-related (estimated) avian mortality was between 16,200 and 59,400 birds in the Southern California region, which was extrapolated to be between 37,800 and 138,600 birds for all USSE facilities across the United States that are either installed or under construction. Importantly, these estimates of direct bird mortality at utility-scale solar energy developments are higher than for wind energy developments, at least in the Southern California area [185].

In the first South African study about animal mortality (birds and mammals) and changes in bird species communities at a 50 MW utility-scale concentrated solar power parabolic trough facility, it was found that birds were much more abundant (141.9 birds km^−1^) and species-rich (51 species) in the surrounding rangeland than in the solar field (1.27 birds km^−1^; 22 species). This is one of the few studies that has also included the evaporation ponds in its assessment, and as the deaths of birds and mammals in these ponds verify, they are important to consider and improve in future [186]. These deaths of birds and mammals were apparently a result of concentrations of chemicals (selenium toxicity), which may either kill via drowning or lead at least to high egg mortality and birth defects in birds using such ponds [187].

Solar power plants, as Kagan et al. (2014) noted [183], also trend towards the degradation or fragmentation of natural areas, especially for birds that are endangered, have limited ranges, and have higher habitat requirements. Yet another phenomenon was the discovery of injured and dead waterbirds (Pelecaniformes, Gaviiformes, and Podicipediformes) at specific solar power plants (SPPs) in the USA, which made scientists hypothesize that birds may perceive large areas of SPPs as water bodies (the “big lake effect”) [187], but this does not seem to occur at every large-scale solar energy facilities [188].

Apart from the effects on birds and some mammals, a recent study found that major solar installations may have very far-reaching and disconcerting implications not just for birds, but for the climate itself. Long et al. (2024) studied a large solar power plant in the Sahara and found that it has an effect on weather patterns in the rest of the world, through massive deployment of solar energy. Thus, a solar plant of large dimensions (depending perhaps also on location) may itself become an agent of changing and disrupting global climate patterns [188].

In 2024, a detailed summary of all forms of solar power installations [189] concluded that impacts on birds and bird mortality depend on a combination of three factors: the location of the SPP, its type, and its size. Negative impacts on avifauna increase sharply with the size and power of USSE projects, their spatial height, and the presence of nearby water bodies. The study proposed that mortality rates can be reduced provided that USSE constructions are placed far from bird migration routes, wetlands, and agricultural lands, and that various technical solutions are implemented, such as acoustic, visual, tactile, and chemosensory means to deter birds from installations, as well as developing mathematical models and improved software for CSP facilities [189]. Some of these recommendations may have come too late, since many installations are found on agricultural land.

#### 2.4.3. Geothermal Power

Alternative energy projects are not confined to wind power and solar energy. There are many other ongoing experiments and schemes concerned with finding or establishing alternative and renewable sources of energy. The most advanced, and perhaps with the longest history, is geothermal energy. Currently, more than 80 countries are involved in geothermal energy production, among them the United States of America, the Philippines, Indonesia, Mexico, Italy, Iceland, New Zealand, and Japan, who produce more than 90% of the world’s geothermal energy [190,191].

The most widespread and highest production of geothermal energy is found in the USA. The Geysers geothermal field in California is the largest geothermal power plant in the world, containing 22 geothermal power plants with a net capacity of 1517 MW [192]. Geothermal power plants are now dotted throughout the west of the United States, and so far, there is also one in Hawaii at Puna and one in Alaska at Chena Hot Springs [193]. The emissions from the Geysers geothermal field spread to considerable distances and apparently cause substantial damage and deaths to California’s long list of native species [194]. The total installed capacity (MWe) from all sources increased by 4.2% over the 5-year period from 2013 to 2018, and annual output increased from 4066 to 4171 TWh/year (+2.6%) [195].

Geothermal energy production in Iceland had wide popular support [196], but there are a multitude of problems, as observed also in New Zealand, such as ongoing subsidence of the land and fluid withdrawal for the power plant [197]. Another study, a master’s thesis on a geothermal power station in Kenya, showed negative impacts on native birds, largely in terms of habitat loss and species disappearing from the area of the power plant [198], and one study of a hydro scheme in the Western Himalayas provided proof of substantial habitat loss for montane birds [199].

Overall, it seems that relatively few studies installing alternative energy plants have commissioned or received post-installation assessments on the impacts on wildlife, let alone birds, and some of the consequences may still be poorly understood. But in all of these various scenarios, we know that harm is done to birds, and it is not difficult to find more examples of the direct mortality of birds [200].

### 2.5. Power Lines

At least for the past two decades, powerlines have become a new and highly relevant issue involving bird collisions and electrocutions, because the generation of electricity from alternative sources requires new powerlines right across all networks [201]. The new technologies of solar, wind, and hydro will require many more thousands of kilometres of new gridlines to supply the various consumers, be they companies, industry, private households, or the increasing number of cars that must be fuelled not with petrol but electricity produced via solar power or other means (Figure 3). The transmission and distribution of power lines are central to the new fossil fuel-free economy. Even the relatively small Western European network of power lines is more than 11.2 million kilometres long [202]. The new energy supply network requires a much expanded and new power line infrastructure. Estimates are that the total grid length will increase by as much as 90% between 2021 and 2050, as published by the International Energy Agency [203]. In this specific sector of new technology, plans for reducing bird mortality at power lines yet to be built are already well advanced [204].

The problem of bird deaths caused by electrocutions on (or collisions with) powerlines has long been known [205,206,207] and has been documented since the 1970s, but it has not been studied in some important geographical areas [208]. Mitigation proposals have been in place since the 1980s. Too often, however, “mitigation” has not been an expression of concern for the safety of the birds but, rather, for preventing outages or costs caused by birds. Birds can indeed cause major outages and damage (e.g., as a result of nest structures on powerline poles), and these can be expensive to repair. For instance, in Florida, monk parakeet nests caused 198 outages in 2001, and as a result, 10,000 consumers had no power [209]. The tactics employed in response tended to be hostile, such as removing nests at or before breeding times, or using deterrents such as installing devices that trigger pulsing lights or deliver small electric shocks.

However, some countries have developed avian protection plans for powerlines. In the USA, Edison Electric, US Fish and Wildlife Services (USFWS), the National Audubon Society, and 10 utility companies formed an Avian Power Line Interaction Committee (APLIC) in 1989 to define bird-related problems and to design ways of reducing bird deaths [210]. The interdisciplinary collaboration led to effective planning and to some actual mitigation that had positive outcomes for both people and birds [211]. For instance, steel poles retrofitted using insulation covers decreased outages by 80% in the USA. In Seattle, nesting platforms with avian-safe clearance were constructed, which reduced bird deaths by 53%. The African–Eurasian Migratory Waterbird Agreement (AEWA) produced a series of very useful and extensive guidelines on a variety of simple methods that have been tested and shown to be successful in avoiding collisions and electrocutions of migratory birds [212]. That was in 2011, but not all countries followed these guidelines [213], and yet the agreement has continued to play an important role internationally [214].

Of course, nesting on power poles is a reflection of the shortfall of appropriate nest sites (trees, nest holes, platforms), and some parrot species and storks have nowhere else to go at breeding time but to use human-made structures [215,216]. Part of any mitigation should therefore be a plan for restoration of habitats. Successful collaboration between bird organisations and the engineering and industry sectors is and remains rare, however; even though individual successes are now a matter of record and there is no lack of advice on the topic, some of the problems of collision and electrocutions related to the scarcity of nest sites and the use of pylons persist [217,218].

As a word of caution on taking the bird fatalities data at face value, the above-cited data on direct mortality in all of the cited cases (from cats, windows, cars, etc., as outlined above) are estimates and, for the most part, are likely to be substantial underestimates. The reasons for this are easily explained and well known to researchers: carcasses of birds tend to disappear rapidly as food for scavengers and a multitude of carnivores, including snakes that swallow carcasses whole [219]. Moreover, if hit by cars, birds (especially small ones) will often leave almost no trace [219,220].

### 2.6. Toxicity and Pollution

#### 2.6.1. Toxins

Then there is also the problem of toxicity. Organic pollutants, including pesticides, have been on the radar for a long time because of their highly toxic bio-accumulative properties and because they can cause various diseases in humans. Actions related to bird poisonings have been less rigorous. The topic of poisonings loomed large, though, in the DDT debates of the 1960s and 1970s [221,222], particularly with respect to raptors [223,224,225,226], leading also to concerns about human health [227] and the effects of pesticides on birds [228], but this had fallen off the public debate once DDT was banned in 1987 in Australia (17 years later than in the USA and most European countries). However, research on contaminants has resurfaced, and the disconcerting results have been vigorously debated again in the last two decades, and particularly in recent years [229].

Neonicotinoids make up one of the most pernicious groups of insecticides that are having a devastating effect on birds and the environment [230,231]. First introduced in the United States in 1994, these pesticides are found in hundreds of products, including insect sprays, seed treatments, soil drenches, tree injections, and veterinary ointments to control fleas in dogs and cats [232]. Neonicotinoids that have received approval for agricultural uses in Australia since 1994 include imidacloprid, clothianidin, thiamethoxam, thiacloprid, acetamiprid, and dinotefuran. Neonicotinoids are banned in the EU because they are toxic to bees, but they have so far not been entirely banned in the UK. Prof. Dave Goulson, a bee expert at the University of Sussex, has been quoted as saying that one teaspoon of these chemicals is enough to kill 1.25 billion honeybees [232]. In 2024, the newly elected UK government made an election promise to completely ban neonicotinoid pesticides (imidacloprid, clothianidin, and thiamethoxam), largely due to their impact on bees. However, British Sugar and the National Farmers’ Union (NFU) have since applied to be allowed to use Cruiser SB, a neonicotinoid that contains thiamethoxam, on sugar beet because it combats a plant disease known as virus yellows by killing the aphid that spreads it [233].

In the USA, a few states have started banning neonicotinoid-treated soy, wheat, and corn seeds in agricultural production. Neonicotinoids have been in use in the USA since 2005, up until research began to show that effects on human nicotinic acetylcholine receptors raise concerns for potential adverse human health impacts even with chronic low-level exposures [234]. However, 6 years before this report on potential effects on humans, a paper was published in *Nature*, titled “Declines in insectivorous birds are associated with high neonicotinoid concentrations” [235]. Several more research papers have further confirmed that the decline in bird biodiversity is also linked to neonicotinoids [236,237], and a report in 2024 analysed the effects of neonicotinoids further and came to the conclusion, as its title indicates, that neonicotinoids impact *all* aspects of bird life [238].

Neonicotinoid insecticides have been the world’s most widely used insecticides until recently, but their use is increasingly blamed for a long list of negative environmental impacts and their direct effects on the health, behaviour, reproduction and, ultimately, survival of birds as they are exposed to polluted food, water, air, or soil. Importantly, one study [239] that used an application rate corresponding to the recommended levels in Spanish regulations found that the recommended dose killed all adult partridges (*Alectoris rufa*) exposed to it within 21 days [239], and a study of the critically endangered Japanese ibis (*Hipponica hippon*) found that even surviving birds experienced problems with reproduction (reproductive dysfunction) when they had ingested neonicotinoid insecticides [240].

Another troubling aspect of neonicotinoid insecticides may be their shelf-life. A study of French farmlands showed that neonicotinoids were present in the soil, with detrimental effects on birds, long after they had been banned some years earlier [241]. One is reminded of 1080 (sodium fluoroacetate), which has been used now for decades in Australia for the deliberate and ongoing poison baiting of dingoes, foxes, and other so-called “nuisance” or invasive species: 1080-laced meatballs can remain active for 8 months on the surface and can kill a number of non-target species, including domestic dogs [242].

However, negative health effects of the described toxins are not the only problem. Herbicides such as dicamba, approved as a replacement for glyphosates in the US and sold as a weed-controlling herbicide, are also having widespread effects on non-target vegetation that birds depend upon, as well as on diverse insect populations—both food sources for birds [243,244,245].

The evidence, as documented here, is overwhelming that we are causing long-term pollution of all of the resources needed by birds, other animals, and humans. Birds, once again, are like the canaries in the mine, and their deaths let us know that humans are at risk from the same contaminants [246]. Soil lead contamination, for instance, is especially devastating for breeding birds. Researchers have shown that soil lead contamination has a negative effect on birds’ ability to produce offspring [247]. Heavy metal pollution in the Pacific Ocean, particularly rising mercury emissions posing an ecotoxicological risk, has been studied extensively in albatross species [248].

#### 2.6.2. Plastics

Contaminants also include plastics, which are now found in large deposits floating in the open oceans, freshwater lakes, and rivers and kill turtles in particular, which mistake the transparent plastics for jellyfish [249], one of their favourite food items [250]. Plastics have also been found in seabirds, such as Laysan (*Phoebastria immutabilis*) and black-footed albatrosses (*Phoebastria nigripes*) [251], and it has been shown that micro- and macroplastics induce multi-organ damage and eventually kill the birds [252]. Estimates suggest that several million tons (19–23 metric tons) of globally generated plastic waste enters aquatic environments every year [253].

Of particular concern are microplastics, which are the remnants of non-degradable plastics that, over time, break into smaller and smaller pieces [254]. Microplastics have accumulated in marine environments and have been shown to harm wetland, shore, and sea birds and, of course, other sea animals [12,255], largely via ingestion and accumulation in their gastrointestinal tracts [256,257]. Microplastics are now also found in terrestrial, freshwater, and so-called “pristine” environments [258,259], and even in the air we breathe [260]. Exposure to microplastics can cause oxidative stress, immunotoxicity, neurotoxicity, reproductive impairment, and endocrine disruption in both terrestrial and aquatic animals. Worse still, microplastics are not the only pollutants found in water. Neonicotinoid insecticides and heavy metals may be found in the same waters and at the same time [235,258,261].

#### 2.6.3. Oil

Richard and colleagues [262] recently identified nine pollutants (pesticides, heavy metals, oil, noise, light, plastic, air pollutants, pharmaceuticals, and radioactive pollution), of which just three made up ¾ of all pollutants, and these three also kill birds: pesticides, heavy metals, and oil [262].

Oil is known as a major pollutant; it is well known that all mining and oil fields have wastewater disposal facilities, and direct avian mortality rates at those open ponds are apparently between 500,000 and 1 million annually. It seems that this practice of open toxic wastewaters (oil pits) has been tolerated for many years [263], although the impact on birds had been recognised, possible mitigations discussed, and some measures put in place decades ago [264]. Relevant here is that fossil fuels and oil fields are still very much in operation, and contamination of the oceans via regular oil dumping (chronic) and accidental major spills continues to the present and continues to kill birds [265,266].

Accidental oil spills may not happen too often, but when they do occur they decimate populations of sea and shore birds, turtles, and other sea-dwelling animals. The most often cited oil spill disaster happened in April 2010, at the oil drilling rig Deepwater Horizon (BP) in the Gulf of Mexico. It was the largest oil spill in the history of marine oil drilling operations to date. Over 4 million barrels of oil flowed into the Gulf. The oil spill from the well lasted for 87 days, before it was finally capped in July 2010. The oil slicks covered over 112 million km^2^ in the upper surface waters. During the spill, researchers counted at least 8500 dead and impaired birds, representing over 93 avian species. In addition, there were more than 3500 birds observed to be visibly oiled [267].

These cases of direct mortality in birds are invariably slow deaths, because their oil-coated plumage does not allow the affected birds to take off in flight or obtain food [268]. Even those individual birds that survived and had only relatively light oiling, as research later showed, were significantly affected in terms of their overall health, with effects on multiple organ systems, cardiac function, and oxidative status. Exposure to oil also especially affected migratory birds from colder climates. Thermoregulation is of critical importance to homeothermic species with high energetic demands, such as migratory birds. Oil spills in colder climates have illustrated the detrimental effects of extensive oil coverage on body contour feathers, resulting in transdermal oil exposure and oral exposure via preening. This has an effect on birds’ survival for three reasons: First, via preening, oil is ingested. Second, oil reduces buoyancy, which, in turn, increases the surface area of the bird exposed to cold water [268]. Third, feather barbules become matted, further reducing the insulative properties of feathers, and this, as well as reduced buoyancy, results in significant heat loss [268,269].

Oil extraction and contamination is not just confined to soil and water but has an effect on the atmosphere and air quality. Crude oil is known to produce high levels of carbon dioxide (one barrel of synthetic crude oil released from oil sands produces 134 kg of carbon dioxide) [270]. A Canadian doctoral thesis from 2014 examined air quality in oil sands (bitumen) deposits via any presence of contaminants in the air and their potential effects on several species of nesting birds near such oil sands sites: Japanese quails (*Coturnix japonica*), American kestrels (*Falco sparverius*), and tree swallows (*Tachycineta bicolor*). The study found significant toxic air contamination (more than 5-fold higher than in control areas) affecting both nestlings and adult birds [271].

Singh and colleagues (2023) were left with just one conclusion, namely, that pollution now plays a major role in avian and, generally, in biodiversity decline [11].

### 2.7. Illegal Hunting and Poaching

Of late, illegal hunting seems to have increased and has come into sharper focus once again [272,273]. Scholars of tropical rainforests have argued over many years, however, that even legal hunting decimates numbers, with substantial consequences for the entire ecosystem [274,275]

Indeed, shooting individual birds, even when permitted, can have disastrous effects on social composition and networks, as well as on reproduction, a consequence rarely mentioned or even considered as a serious threat to maintaining numbers. The effects of hunting are not just measurable in the number of birds that have been killed but should also include those that are injured and left in situ, as happens in duck shooting. Duck shooting is a highly contentious if not controversial issue in Australia and some other countries [276]. There are many species that are monogamous, and some of these, at least in Australia, are classified as vulnerable or endangered. Shooting “for fun” in such cases would appear to be so entirely against the spirit of conservation that, arguably, this practice should not just be a matter of regulation but phased out completely and, preferably, for all birds.

As Wilkie and colleagues argued over a decade ago,

“Hunting is an insidious but significant driver of tropical forest defaunation, risking cascading changes in forest plant and animal composition. Ineffective legislation and enforcement along with a failure of decision makers to address the threats of hunting is fanning the fire of a tropical forest extinction crisis” [277].

The tropical forest extinction crisis that Wilkie and colleagues referred to also relates to the “empty forest” syndrome. This term “empty forest”, coined by Redford (1992) [278], largely referred to overhunting but has now been broadened in the literature to “defaunation”, which includes all threats to and deaths of fauna by any means [279].

Studies in the UK show that avian species can recover from past downturns in population, as they did after the impacts of organochlorine pesticides [280], when changes in legislation and/or imposed penalties are implemented resolutely. Appropriate laws can regulate or remove and ban pesticides. In 2020, a paper revealed that, in the European Union alone, over 80 million wild birds are shot annually to supply about 1% of the EU population (448 million). What is less well known is that hunting and clay target shooting use lead-based gunshot, wads, and bullets, which are then scattered and stay in the environment. Annually, this amounts to over 40,000 tonnes of un-reclaimed lead (a major pollutant) that is dropped into the environment (often into wetlands) of EU countries. Arguments even just about the transition to lead-free ammunition have taken years [281]. A ban on the “right” to shoot wild, native, and migratory birds is not likely to happen soon, partially because it is legitimised in the meat/game market as a sellable product and persists there despite significantly changing community attitudes.

Poaching, just like hunting, is a wilful and violent act of removing an individual bird from its social group and native environment. It was once a major topic, which led to import and export restrictions [282]. However, it is anything but a matter of the past [283]. Poaching continues to be documented as thriving in Africa [284,285], Central America [286,287], South America [288,289], Southeast Asia [290,291,292], Europe [293,294,295], the USA [296], and Australia [297].

The illegal wildlife trade is now the second-largest black market worldwide, after narcotics [283], partly because of fusions of trafficking in drugs and wildlife [298]. Poaching usually means that the hunters want to derive profit from live animals. However, when birds are captured from the wild and sold across borders, they have to be hidden like contraband [299]. The fatality rate from smuggling parrots was, and presumably still is, extraordinarily high. In Mexico, an estimated 75% of parrots (50,000–60,000 bird casualties) that were poached died in transit from their natural habitat to their black market destination [300,301].

Spoon concluded in 2006, concerning parrots, that ”the single-most threat to natural populations is the capture of individuals for aviculture and the pet market” [302]. Greg Warchol noted twenty years ago [303] that the trade in birds is better organised than many other branches of the illegal market in wildlife, specialised and dominated by collectors, breeders, and dealers in rare and exotic species. They rely on verbal agreements, internet sites, and even classified advertisements in bird enthusiast magazines to order, sell, or trade birds [303]. Hence, there is a hidden human activity that is a significant trauma for birds, be they migratory songbirds or parrots of colourful plumage or significant size 

Importantly, this illicit trade indicates another (and often brutal) way of reducing bird numbers in their natural habitat. Since poaching has remained a substantial and illegal business, not just in poor countries (subsistence poaching) but also in wealthy industrialised countries (exclusively profit motive), one should expect more concerted and consistent actions in this field [304].

Are there solutions? Countering illegal trading activities in wild birds will continue to require more funds to monitor, identify, and reduce the number of poachers. There are already many existing environmental laws that, however, tend to be partially ineffective, at least in some countries, be this for local reasons, for lack of funds to employ an appropriate number of wildlife officers, or because existing laws have loopholes, are too general, and are not strongly enforced or even enforceable [305]. If individuals are caught and actually prosecuted, they may be asked to pay fines that may just be a fraction of the monetary value that such illegal trade can bring [306]. Avian traders are seldom deterred by the relatively low maximum fines for trafficking, since the value of the birds may exceed the value of the fine. In one study, an officer was quoted as saying that “there is more profit for smugglers in exotic birds than there is in cocaine” [307].

Indeed, Stokes (2006) [308] made the important point that species’ risk of extinction, particularly in birds (parrots, songbirds), is also subject to market forces according to what humans like to buy and own. These, in turn, are decided by people who may know absolutely nothing about conservation and, most objectionably, may have no knowledge of, insight into, or even care about animal species that they like to shoot, acquire, keep, or sell. Worse still is that they are being listened to and may determine marketing strategies and even conservation priorities. Implicit in this illegal trade, and in shooting for fun, is the notion that wildlife, including birds, is ab ovo regarded as property that can be caught, bought and sold, and even mistreated and neglected without notable repercussions [303,305].

Having summarised the major causes of direct mortality in birds, it is easy to see that the concatenation of an unprecedented number of deadly anthropogenic factors (actions, structures, substances, and technologies) acting at the same time in the real world can amount to billions of birds lost each year. One would think that it is now more than urgent to take every opportunity to slow down or halt what is now openly called the “sixth mass extinction” [309,310,311,312].

## 3. Section III—Direct Consequences of and Possible Reductions in Direct Mortality

### 3.1. Effects of Direct Mortality on Surviving Individuals/Populations

The direct causes of mortality detailed above occur in a context of habitat loss, climate change, and sea level changes. Climate and sea level changes, however, have occurred throughout the history of the Earth and, apart from the well-studied ice ages, one climate change (post-ice age) has been documented as recently as the Holocene [313]. Recording another climate change is in itself not necessarily a problem [314,315]. The modern problem is the speed with which these changes are occurring [316]: not over tens of thousands or millions of years, but over a few hundred years or even just decades [317,318]. In addition, the invention of technologies, such as for human comfort (buildings, cars, and countless other conveniences for humans to live comfortably), and now a new layer of technologies and innovations to fight rapid climate change (wind turbines, solar panels, geothermal and hydro power plants, and even drones), has created very dramatic conditions and often insurmountable problems for birds. Caro and colleagues (2022) said it very well in the title of their paper: “An inconvenient misconception: Climate change is not the principal driver of biodiversity loss’ but rather habitat loss and overexploitation” [319]. Also, the way deaths or delayed deaths and injuries are caused alters the prospects for future avian success. One problem is reduced fitness. Nemes and colleagues (2023) gave the example of migratory birds and argued the following:

“Many migrants are killed due to encounters with artificial light, introduced species, pollutants, and other anthropogenic hazards, while survivors of these encounters can suffer longer-lasting negative effects…Building collisions frequently kill migrating birds, but the numbers of migrants that survive with an impaired ability to fly, refuel, or navigate to their destination on time is not well understood. Though not immediately fatal, such injuries can lead to delayed mortality and, ultimately, reduced lifetime reproductive success. Furthermore, migrants are likely to encounter multiple threats on their journeys, which can interact synergistically to further reduce fitness. For instance, light pollution attracts and disorients migrants, increasing the likelihood of window strikes, and surviving birds may be more vulnerable to predation from introduced predators.” [320].

These are important observations. However, survivors are compromised by more than a concatenation of events external to them. Nemes and colleagues also argued that a full understanding of the impacts of human activity on birds “must include the cumulative and interacting effects that extend beyond immediate mortality” [320].

Indeed, there are interacting and additional factors that are undermining and indirectly affecting mortality, or at least lead to poor performances in breeding. As said before, unlike most other animals, humans and birds have in common that most of them live in nuclear families (parents and offspring) and some in extended families. In birds, at least 95–97 percent of more than 10,000 avian species pair-bond, however briefly, and jointly raise their offspring [321]. How exceptional this is can be gleaned from the fact that this is very rare across all extant species and found sparingly even amongst mammals: only about five percent of mammals, including primates, are said to form lasting, let alone lifelong, pair bonds or even short-term pair bonds and raise offspring together [322].

Stable long-term social bonds have indeed a number of substantial advantages, particularly relating to the survival of a high number of offspring and raising them to produce viable offspring in turn.

The studies that we do have suggest the following about pair bond formation:(a)It is based on choice, as we know that force-paired pairs in the pet market often will not breed or do poorly at reproducing [323];(b)It is based on pre-sexual maturity bonding in some cases [324];(c)It tends to be assortative [325], although the importance of this may have been overstated [325], but “familiarity” with a partner seems to be associated with highly successful breeding: having larger clutches, higher hatching, and fledging success [326];(d)Individuals in pair-bonding birds have prosocial skills that tend to lead to cooperative and cognitively complex behaviour [90,326,327,328,329];(e)Lifelong partners are more successful in reproduction and offspring fitness [330];(f)It has long been recognised that intact avian families have the best chance of survival, partially because of increased parental investment in well-matched pairs [330,331].

Birds with long-term socially monogamous relationships have been studied in detail, and they belong to a wide range of diverse families and orders, including waterfowl such as water and shore birds, oystercatchers (*Haematopus ostralegus*) [332], and greylag geese (*Anser anser*) [333], songbirds, such as Steller’s jay (*Cyanocitta stelleri*) [334], tit species (Paridae) [335,336], zebra finches (*Taeniopygia castanotis*) [337,338,339], the bearded reedling (*Panurus biarmicus*) [340], and many others; and many parrot species (Psittaciformes) [341], particularly budgerigars (*Melopsittacus undulatus*) [342] and cockatiels (*Nymphicus hollandicus*) [343], as well as corvids, such as common ravens (*Corvus corax*) [344,345] and jackdaws (*Corvus monedula*) [346], displaying social behaviour that has also been linked to highly developed cognitive abilities [88,89].

Research has also shown repeatedly that pair fidelity results in increased breeding success, larger numbers of surviving offspring, and greater survival of pair members [335]. Such reproductive success has been related to mate familiarity, usually observable as improved coordination and cooperation in pair behaviour [329]. Veltman and Carrick (1990) [346] clearly showed that there is a direct link between the survival of fledged offspring and time spent in the natal territory under the protection of their parents/social group, as tested in Australian magpies (*Gymnorhina tibicen*): of those that left before the 1st year post-fledging (which is the norm [347]), not quite 50% survived; when staying for two years, survival increased to 70%; and for those fledglings that had stayed for 4 years (very rare), the survival rate was nearly 100% [347,348]. These data closely align with the findings of another study in Central America, on the western slaty antshrike (*Thamnophilus atrinucha*), where nearly half of fledglings died in the first weeks of independence [349].

Knowing the advantages that the stability of such social arrangements bestows [350] raises questions about the effects for survivors of anthropogenic causes of mortality and what this may mean in terms of the decline of birds. The repercussions of direct mortality for the social unit of the deceased are often overlooked and rarely, if ever, discussed in the animal context.

However, sudden deaths in human families have been subject to very extensive research [351,352], and since most birds have very similar family structures to humans, it may also be crucial to investigate these effects in birds in future, and to ask how such sudden deaths contribute to the dysfunction and decline of avian family units and even species.

Moreover, poaching, shootings, and indeed any fatal anthropogenic incidents take individual birds *indiscriminately*, be they inexperienced juveniles or experienced adult individuals, the very healthy and successful, long-lived and bonded birds, or fledglings that are just beginning to explore their environment.

It is not difficult to conclude, or at least to hypothesise, that such indiscriminate direct mortality may well also be highly socially disruptive for birds, and especially for offspring, be they nestlings or juveniles. Juveniles have to learn a great deal during their development, including relating socially and vocally to their parents or larger social group [353,354]. The decisions that they might have to make in future are dependent on these early months of life [355]. Depriving juveniles of developmentally important experiences has been experimentally confirmed as having negative effects on spatial learning and memory in juveniles [356].

For adults, the impact may be of a different kind. When a partner is killed, such a loss is likely to be particularly stressful. We do not know whether bonded birds grieve, as do humans, but the disappearance of a partner may, for instance, induce waiting and vigilance behaviour as if the partner might return, especially if a death has not been witnessed [357]. The death of a single bird will certainly have detrimental effects on the social groups or the few remaining family members and, when applicable, could have deeply disruptive effects on reproduction [358]. A surviving adult bird may not seek another partner for some time to come. This may vary with species, of course. Some species may find another partner during the next breeding season, such as the Eurasian bullfinch (*Pyrrhula pyrrhula*) [359] while the female tawny frogmouth (*Podargus strigoides*) may reject future suitors for several years after a partner’s loss [360].

Hence, when bird deaths/direct mortality figures are listed, these estimates cannot take into account the hidden effects. Indeed, more often than not, an entire family has been affected, with potential long-term negative consequences for that social group. At the very least, such deaths introduce instability, increase vulnerability, and lead to the possible death of nestlings, and even to the abandonment of home ranges or territories. Of course, the latter can occur for very different reasons, such as forest fragmentation, which often results in a catastrophic decline in nesting success [361]. The point here is that a significant number of such negative effects do not show up in statistics on direct mortality, but they may well be significant contributions to bird decline, and these factors themselves are attributable to anthropogenic factors, as detailed above, which have caused direct mortality.

A final point that is rarely raised is that breeding season does not mean that every adult gets a chance to breed. Conditions may not be right, be this because of adverse weather conditions, sex bias in potential partnerships (e.g., too many males vying for far fewer females, or vice versa), or the possibility that a bonded partner may not return after a year of separation, as in albatross or penguins. In some mating systems, such as lek systems or facultative social polygyny, some males mate with many different females at the expense of other males, leaving some or even many males unable to mate at all [362]. In territorial birds, individuals may not have found a suitable territory and may remain itinerant for most of their lives, without a chance to establish a secure base for raising young [362,363]. Hence, failures in the breeding season can have far more serious consequences for a species as a whole than even the statistics on direct mortality suggest.

In the Anthropocene, one is well aware that declines in bird numbers and species disappearances from specific environments can now happen quite swiftly, be this for genetic, ecological, or anthropogenic reasons. Stojanovic and colleagues (2022) [364] reminded us that, for genetic reasons, lineage loss at small population sizes “can act as a stochastic process, affecting population genetic structure and extinction risk. Lineage loss may occur rapidly, with more than 80% of lineages lost in some mammals and birds over only two decades” [364]. Rigal et al. (2022) [365] noted the changes in and loss of biodiversity, commonly assessed in terms of changes in species distributions and community composition. However, as the authors noted, local dynamics may change dramatically if a group of birds leaves or if just one or two species are removed via external processes. This process can destabilise local assemblies. Indeed, as Rigal et al. argued, replacement of a diversity of mainly specialist avian species by a few generalists, triggered by ongoing global change, is “considered as one of the most pervasive aspects of the biodiversity crisis” [365].

In summary, the repercussions of direct and indiscriminate mortality ripple through the entire living world, unseen and often unrecorded, adding to the actual recorded and estimated figures of direct mortality and decline among birds. Taking these secondary effects and, for birds, often disorienting factors into account also confirms that the damage to birds goes much deeper and is far more destructive and more widespread than the direct mortality figures suggest. We need to comprehend how many anthropogenic stressors have been added to, and act simultaneously on, the environment and its wild inhabitants (here, birds).

### 3.2. The Human Response: Ignorance, Impotence, or Indifference to Act?

This question is not rhetorical. We know that the current loss of bird numbers and species (which is accelerating) is unsustainable [39,242,366,367,368,369]. We know this because at no time in human history have so many worldwide tools been developed for international assessments of the state of the Earth. There are tools and indicators for assessing and benchmarking environmental impacts, such as the Strategic Environmental Assessment (SEA), Environmental Impact Assessment (EIA), Environmental Risk Assessment (ERA), Cost–Benefit Analysis (CBA), Material Flow Analysis (MFA), and ecological footprint, and even Life Cycle Assessments (LCAs) for industry, annual reviews of environment and resources, and the pronunciation of a range of international treaties expressed in a number of international conventions and publications.

Even more impressively, at no other time in human history have so many international bodies been established post-WWII to monitor and provide guidelines and information on climate, animals, plants, and resources. There are the so-called “Big Five” global conservation treaties—the Ramsar Convention on Wetlands of International Importance Especially as Waterfowl Habitat (1971), the World Heritage Convention (1972), the Convention on International Trade in Endangered Species of Wild Fauna and Flora (CITES, 1973), the Convention on the Conservation of Migratory Species of Wild Animals (CMS, 1979), and the Convention on Biological Diversity (CBD, 1992). Then there are the United Nations Environment Programme (UN Environment, 1972), the World Meteorological Organization (WMO, 1950), and the Intergovernmental Panel on Climate Change (IPCC, 1988), starting with 195 member countries. The World Wildlife Fund for Nature (WWF) was founded in 1961, and shortly thereafter, in 1964, the IUCN (International Union for Conservation of Nature) was founded with the brief to establish an inventory of the general conservation status of all living species. As is well known, the IUCN created the infamous “Red List”—a dramatic record, even if incomplete, of the accelerating decline of species generally. In recent times, a multitude of charities have sprung up all over the world, some with briefs specific to particular species (such as for tigers or great apes), but also many worldwide organisations for animals and their welfare in general, and some specifically for birds (such as BirdLife International and countless ornithological and other volunteer organisations and institutions, such as the Cornell University Laboratory of Ornithology) that have made a major impact via their publications.

As a result, the last few decades have produced a remarkable body of information on the state of the world during climate change, and as far as animals and the natural environment are the centre of concern here, the reports and opinion papers have been very comprehensive. They appear regularly and report on a global scale, including the status of birds.

These reports could not be more pessimistic. A book by Nemetz (2022) even asked in the title “are we losing the battle to save our planet?” [367]. Publications such as the “Living Planet Report” of 2024 [4], or reports on the status of birds [368,369], paint a very gloomy picture indeed of the downward trend of all living things, from invertebrates in the soil and water to fishes, birds, and mammals. All categories have shown a downward trend of an average of 68% since 1970, and as far as we know and can judge from these many published figures, among vertebrates, birds have perhaps shown the steepest decline in numbers (see Figure 1 above).

Forums and conservation movements have rightly wondered whether being in possession of all this information has resulted in meaningful changes. Donaldson and Kymlicka (2013) [370], for instance, maintained that animal advocacy groups had “failed to reduce the numbers of animals killed, consumed, and failed to create meaningful legislation”. Sorenson argued that all of this engagement with animals over decades has even failed “to change fundamental attitudes towards animals” [371], and as far as predictions go on climate change, greenhouse gas emissions are expected to continue to increase, even substantially, and this has to do with rising populations, increased food needs, and changing/accelerating demands for red meat. As Tilman and Clark (2014) calculated,

“By 2050 these dietary trends, if unchecked, would be a major contributor to an estimated 80 per cent increase in global agricultural greenhouse gas emissions from food production and to global land clearing” [372].

Crist (2016) argued powerfully how urgent it has now become that we must choose a “planet of life”, showing that an ideology that promotes endless economic growth is one of the most significant causes of unsustainability resulting in the disappearance of habitats and species [373]. Another culprit, according to Crist and Cafaro (2012), is overpopulation [374]. Cowie and colleagues (2022) [312] fear that most of the Earth’s biodiversity will disappear, and much of it (especially invertebrates) without a trace. Denying the crisis, so they write, and “accepting it and doing nothing about it, or embracing it and manipulating it for the fickle benefit of people, defined no doubt by politicians and business interests, is an abrogation of moral responsibility” [312].

In this paper, the most obvious cases of direct mortality of birds from anthropogenic sources have been discussed in great detail. Many of the rarer—but at times also deadly or in some way debilitating—consequences of human actions on birds have not been included here for various reasons: either because they are event-dependent and do not happen often, or because we do not have enough data yet.

However, it is important to mention that an ever-increasing stream of technological innovations may have implications for birds’ survival. This includes new technologies that are directly employed in bird studies, the use of which is steadily increasing. Such technology includes the latest generations of GPS trackers (telemetry) [375] and drones, both pieces of technology used for bird studies to study their decline or breeding success that may ironically work against birds themselves. The harm that GPS trackers can do to birds was explored in a recent paper [149], and there is now evidence that using drones close to birds and bird nests may harm birds or their breeding processes [376]. In fact, Borrelle and Fletcher noted that the use of drones can lead to breeding failure [377]. My own research attempts with a drone some years ago were terminated when birds shrieked in fear, were flushed from trees, and a pair of black-faced monarchs (*Monarcha melanopsis*) abandoned their nest and territory. No other pair of black-faced monarchs has ever been noted breeding again in this 200-acre sub-tropical rainforest and eucalypt woodland location in the following years.

The latest and most comprehensive summary of publications on the effects of drones on birds has just been published in 2024 [378], and there is no need to duplicate their findings here. However, the authors of this recent study concluded that drones should be considered invasive and that signs of adverse behaviours in birds should be noted and recorded. These behaviours might include intense levels of alarm calling, vigilance behaviour, and/or nest abandonment [378].

Human fascination with technologies has led people to often uncritically design and adopt many new gadgets [379]. Electronic devices and drones have made life easier for researchers, even though, contradictorily, they may add more stress and hardship to target birds. Human innovativeness has now spawned another product: the flying car, which might be launched in 2025 in various countries. Indeed, there is an entire industry devoted to manned and unmanned aerial vehicles (UAVs), “drone” technologies for surveillance, and package deliveries via flying devices. There is also a growing interest in electric vertical take-off and landing (VTOL) aircraft. An article on the capabilities of the latter was published in 2019 in the journal *Nature Communications,* surprisingly without a single reference to any effect on nature generally or birds specifically [380]; the paper discusses the technical merits of VTOL aircraft. These aircraft, according to Figure 1 in their aforementioned paper, would take off vertically and then cruise at a height of 305 m and at a speed of 241 km/h. Moreover, in Figure 2 of the same paper, a model aircraft is shown, with eight wing propellers (four on each wing) and one propeller on the rear. The word “sustainability” is used in the technical sense of primary energy use and greenhouse gas emissions [380], not sustainable in the sense of becoming (inevitably) a major threat to birds. These technical details make a mockery of all of the research on migratory birds (avian flight corridors) and on known flight behaviour in birds, as discussed above.

Even manufacturers of drones—or as they prefer to call them, ”unmanned aerial vehicles” (UAVs)—have realised that some species of bird, particularly large eagles, will attack such invaders of the sky (flight height of about 125 m, with speeds of up to 92 km/h) and, depending on type, can suffer severe injuries or worse, with exposed propellers potentially amputating their legs as they try to tackle the object with their feet. Alternatively, if the drones do not have such propellers, it is the eagle that will take them down, as was reported in 2016 from the West Australian Goldfields, in which nine UAVs were completely destroyed [381].

Another paper, presenting different designs [382], contains a section titled “Ethical Issues”, in which the writers admit that the operation space of such flying cars is

“the main living space [of] birds, […and…] *irresponsible and wanton operation of flying cars will inevitably negatively impact birds by disrupting their nests, provoking attacks, scattering eggs, interrupting feeding, and midair collisions*. Furthermore, the noise generated by frequent TOL* and flying will also impact ground animals. There are several ways that the negative effects of FCTS* can be reduced, such as by monitoring sensitive areas, poaching records, habitat preservation, etc. An environmental impact assessment should be considered during the design and updating process of flying routines”. *(TOL = take-off and landing; FCTS = flying car transportation systems); emphases added in italics [382].

How the “negative effects” can be reduced is not spelled out other than to say that Environmental Impact Statements (EISs) should be ”considered” (not sought). Unfortunately, there have been too many instances in which such EIS documents, paid for handsomely, said only what their customers ordered. The fact remains that we know in advance that all of the negative impacts on birds in a scenario of wide acceptance of “flying cars” will do exactly what the authors cited above predict. This is yet another invasive technology that will be added to the natural environment (the air/sky) and clearly interfere with birds.

Even at this critical stage of steep decline of birds, humans are generating new collision courses between business interests on the one hand and the urgent call for the preservation of biodiversity on the other. This is not to say that one cannot see very useful functions for such an invention for rescue services, supplying emergency materials to towns that have been cut off by landslides and/or floods, or rescuing and transporting people to hospitals in larger numbers and more safely than helicopters can, but such functions have not been mentioned because the target market is in transportation of goods (large commercial enterprises).

Notable in these developments is that humans have moved into the only space and free frontier for birds: the air. Just half a century ago, the sky was largely the sole domain of birds (at least at the typical avian flight corridor altitudes), and many of these technological changes and innovations are not occasioned by climate change or excusable as necessary changes to slow down extinctions, restore and conserve nature, or aid the survival of birds. On the contrary, these new technologies create additional layers of harm.

It started with high buildings and reflective windows that, as discussed earlier (Section II), have killed millions of birds since, followed by drones, and now flying cars and small electric aircraft, be they hybrid (DA36 E-Star) or fully electric-powered aircraft (Velis Electro). These new classes of small, fully electric aircraft are targeted specifically for missions in and around major urban metropolitan areas, commonly referred to as Urban Air Mobility (UAM). As Courtin and Hansman pointed out in 2018 [383],

“The hazard of bird strikes, whether to the propulsors or cockpit, may be different for electric aircraft and UAM vehicles in particular because much more of the flight is expected to take place at low altitudes where birds are prevalent and because rotors may be significantly smaller. This, combined with flight at relatively high speeds, *may cause the probability or consequence of bird strikes to increase.*” (emphasis added) [383].

These innovations fail any test of progress (improvements without increasing or doing harm to other living things). It is much easier to declare native birds as “pests” when they interfere with human dominance. How likely is it that these airborne technologies will not be allowed to fly during the breeding season or during peak migration periods of birds? Protective regulations should come into existence before these developments appear in the landscape, of course.

One ought to ask why there are not more nationwide coherent responses to save birds. What is less well known or agreed upon is that some of the worst declines could be stopped almost immediately and relatively easily.

So why is the evidence of billions birds dying every single year from anthropogenic causes not evoking an outrage and resulting in demands for immediate ameliorative action at the population and political levels? Cameron [384] called this “compassion collapse” because numbers make us numb and do not evoke compassion or empathy because they are abstract, relate to unknown individuals, and go beyond personal comprehension [384], whereas the presentation of the suffering of an individual, preferably a child, will spark an immediate outpouring of empathy [385]. Asking for compassion for embattled and dying birds may thus not be the most effective strategy, and other solutions have to be found.

### 3.3. Pragmatic Solutions to “Small” Problems with Very High Avian Death Rates

Ironically, the current three most numerically devastating anthropogenic causes of avian deaths may be among the easiest to deal with and perhaps even fix. Any interventions could happen quite quickly and relatively cheaply. For instance, cat kills of birds, scandalously high across the world, could be reduced by one piece of legislation: to confine pet cats to properties (and cat runs, if outside). Interestingly, as Dauphiné and Cooper pointed out in 2009 [386], the legal status of domestic cats should pose no problem at all in curtailing their movements. They said the following of domestic pet cats (emphasis added):

“because they form a domestic species distinct from their wild ancestral species, domestic cats are considered to be an exotic, or non-native, species in all environments in which they occur.” [386] (citing O’Brien and Johnson 2007 [387])

If this categorisation of pet cats still applies in 2024 and in all environments, it is even more surprising to find such shyness in implementing pet cat confinement, given the rigour for removal and extermination attempts with which “invasive” species are generally met. Since we have made an exotic, invasive species a much-loved domestic animal, there seems ample room for a liveable and benign compromise.

Furthermore, there is clearly an awareness of a major problem of fatal bird collisions with light structures [388], and especially with windows [130,131,132,133]. Graham Martin, a specialist in avian vision, explained many years ago why birds may collide with windows head-on and at full speed [389]. Birds may be blinded by sun reflections, but they may also collide with structures in their flight path under favourable light conditions. Birds have laterally placed eyes; hence, their binocular vision is limited and usually at low resolution at a distance, meaning that they have reduced ability to see things directly ahead when flying. They tend to look down or up in the lateral field for orientation and, of course, in their evolution, would not have expected obstacles straight ahead in their flight path/in the sky [389].

One chief reason for collisions with windows is the reflection of the environment in glass windows and glass walls. If the window reflects views of the sky and tree branches, birds, flying at speeds of 25 km/hr or more, may not adjust their flight speed or direction. Furthermore, small songbirds tend to fly in between trees and branches for protection, but, broadly, birds have typical fly zones at a height of some 20–150 m (65–500 feet). Migratory species may fly at 600–1500 m (2000 to 5000 feet), where they can use the prevailing winds to assist them in long-distance flight; a bird may begin to travel at about 1500 m and eventually climb to 6000–8000 m (20,000–29,000 feet) [390]. Importantly, the typical fly zone for birds (local birds and migratory birds) is thus between 20 and 1500 m. This space is now severely contested by cities with buildings more than twelve storeys high, as rising populations move accommodation expansion upwards, with skyscrapers in most major cities around the world, as well as by wind turbines.

A multitude of design possibilities are now available and, in many ways, window science and architecture have been very innovative. Some university campuses have achieved marked reductions in bird losses from window collisions, but some amelioration has been achieved largely by student activities exerting pressure on campus administrations [127,131]. However, the efforts seem to be few and far between—be this in terms of legislation, marketing, actual use, or design of glass or technology—to consider designs with proven ability to avoid window collisions by birds.

Glass windows have been in a state of experimentation for some years, and there are changes to glass surfaces or inbuilt parts in glass that actually work and can reduce the death rate substantially. In the US, copious studies a decade ago showed which buildings, which glass types, and in which directions constituted particular death traps for birds, and the debate spilled over from the ornithological literature to architecture. Such concerns are increasingly recognised in building design guidelines [135]. However, as Brown et al. [169] pointed out in 2016, these guidelines are not legally binding and are not particularly concerned with bird deaths. Ogden argued in 2014 [391] that even the most advanced “green” watchdog of architecture (the “Leadership in Energy and Environmental Design”, or LEED) awarded very few points to solving bird/window collision problems and was largely concerned with energy efficiency and other green goals at the expense of bird death rates [390].

There are now websites that suggest remedies [392]. In some local cases with high night migration of birds, turning the lights off in buildings has helped [393]. In New York, the mayor declared in 2019 that glass skyscrapers “have no place on our Earth” as part of the New York City “Green New Deal” [392], but instead of embracing the change, the architectural fraternity and real estate agencies offered outraged resistance. So far, almost none of the efforts to save birds from such preventable deaths as collisions with windows have tended to occur nationwide. With some simple retrofitting and a set of international building standards, the lives of millions of birds could be saved.

Car collisions with wildlife have received constant attention, and many road safety features have been introduced, such as fencing structures to prevent moose, deer, bears, koalas, or kangaroos, for instance, from running onto the road and endangering cars and their own lives [394,395]. Importantly, the behaviour of wildlife has been studied in detail to make any changes fit the target species [393,396].

However, bird collisions with cars have apparently spawned relatively few ideas on how to reduce the bird casualties. The estimated numbers calculated for some European countries have clearly risen over the years, presumably also related to the number of cars on the road, the size of the roads, and the speed limits imposed. In the 1980s, annual bird casualties per annum were 2.5 million in England, ca. 9.4 million in Germany, and 3.2 million in Denmark, and over 1 million in Sweden, while in Bulgaria in the late 1970s about 7 million birds were killed [397]. Suggestions for making roads more secure have largely centred on planting hedges or medium-sized trees on both sides of the road to enable birds to fly over and above cars and trucks. The very high incidence of road accidents suggests that birds have mastered little to no adaptation to risks associated with cars, however. There has been one study that tested ravens on distance and speed perception of approaching cars and found that, in the majority of cases, they tend to be able to evade oncoming cars just in time [398].

Attention has also been paid to bird strikes of planes, usually by migratory birds [399,400]. In Australia, freeways have been built with underpasses for wildlife or high bridges—either solid bridges planted with shrubs and even small trees, or narrow Hessian structures high above freeways for small mammals such as koalas and possums [142,401]. The emphasis has been on minimising risk to human lives and material damage to structures and vehicles. In such cases, substantial funds have been invested to find solutions that protect humans and lower the damage bill. It does not make sense to make no changes and fail to reduce the death rate in birds in other contexts when there are simple strategies available that cost little.

Clearly, there are many complex problems in the world today that will be difficult to solve. For instance, the same report by Ogden (2014), cited above [391], also suggests that “conservation is critical but not enough–we must also transform food production and consumption patterns”. Of course, this is well known [402,403]. According to Tilman and Clark (2014), too many traditional diets are replaced by refined sugars, refined fats, oils, and meat [372]. They argue that if present trends continue, such trends, particularly beef, would be a “major contributor to an estimated 80 per cent increase in global agricultural greenhouse gas emissions from food production and global land clearing” [372].

Land clearing equals habitat loss for native animals and is a significant source of death rates across the world, aptly titled in one paper as “the invisible harm” [404]. How many native birds are killed by clear felling is not known. It is often presumed that birds fly away when there are catastrophic events such as forest and grass fires or large-scale clear felling programmes of forested areas. The assumption is that birds will find another place to live, but this is not always so when few if any suitable alternatives may be available [404,405].

Some of the most hopeful signs for a change in global attitudes actually seem to arise with the help of the legal professions. Cerullo and colleagues pointed out recently [406] that while the EU has old forest protection policies in place, it has shifted the problem elsewhere because its regulations failed to implement additional rules that limit the importation of timbers. Such failures in policymaking, in turn, have apparently led to the dramatically fast deforestation of the Amazon. These dynamics of developed nations shifting the problem elsewhere have occurred on a global scale. Forests, after all, are tied to a wide range of globally traded products [407]. One could draw the conclusion that cross-disciplinary collaboration (also with other professions, such as engineering) may get us there sooner to stop massive deforestation and defaunation and minimise the threat of further extinctions of birds. It also seems clear from the above that, sometimes, several actions have to be undertaken simultaneously. Regulations may have to come into play that require consultation and discussions with other stakeholders regarding innovation, for instance, in the building industry. Having more, new, and better laws brings with them the question of the rate, manner, and mechanisms of enforcement, i.e., of discovering and dealing with non-compliance.

For instance, the problem with controlling domestic cats (keeping them confined to homes and outdoor cat runs), as Trouwborst et al. (2020) discovered and confirmed [408], is more problematic than it might appear, even if, at face value, an appropriate law should be easy to enforce. The reason for this is that numerous legal obligations relevant to free-ranging domestic cats *already apply* under global treaties such as the “Convention on Biological Diversity”, “Convention on Migratory Species”, and “World Heritage Convention”, along with a range of regional legal instruments for biodiversity conservation. According to their research,

“Many national authorities around the world are currently required, under international law, to adopt and implement policies aimed at preventing, reducing or eliminating the biodiversity impacts of free-ranging domestic cats, in particular by (a) removing feral and other unowned cats from the landscape to the greatest extent possible and (b) restricting the outdoor access of owned cats” [408].

The authors more than hinted that there is little to no justification for non-compliance with international wildlife law. So, perhaps, the power of these international bodies is not strong enough to insist on annual progress reports from those countries that are signatories to such international laws who continue to say “yes” to change but, in fact, do little to nothing, and one suspects that the follow-up of new regulations may be too weak or even non-existent.

In the study on French farmland (2023) discussed above [241], it was clearly shown that neonicotinoids were present in blood samples from raptor chicks (the Montagu’s harrier, *Circus pygargus*). Indeed, residues of five neonics were found in the soil, of which three had been banned in France since 2018—clothianidin, thiacloprid, and thiamethoxam—and two (dinotefuran and nitenpyram) used for veterinary purposes only, and these residues were at higher concentrations than found elsewhere before the ban in 2018. The question arises here whether this is evidence of long-term persistence of the substances or blatant disregard of the bans imposed, especially if there is no testing implemented after such a ban has been introduced [241]. Perhaps it also needs pers ion that the survival of wildlife, including any insects, plants, and birds, is not a marginal issue.

Hence, it would be naïve to believe that simply writing new laws would make problems disappear, unless they are enforceable and, as said before, have consequences for those who do not adhere to such laws. The examples of entirely inadequate fines for poachers and illegal hunters, for example, are discussed above. The fight against classes of pesticides and herbicides has been conducted in court rooms for some time, while the damage continues. Issues of enforcement of the law—of laws that reflect the criminality of any practices that help destroy birds and bats, or that harm or kill pollinating insects—should be foremost agenda items for the welfare and survival of both humans and animals. There are depressing signs of ongoing illegal activities [409], but also inspirational ideas and actions being suggested or implemented right now [410,411,412,413].

## 4. Concluding Remarks

From all of the evidence that we have so far, it is rather clear that direct mortality has repercussions well beyond a single death. A bird may have been hit by a car, been shot by a hunter, been poisoned, or come to grief at wind turbines, solar installations, or geothermal plants. These fatalities do not remain isolated events but are multiplied into thousands and even millions of such individual losses every single day and year, *and* these losses are rising. The details of direct mortality and their consequences described here are indeed a blueprint for the mass extinction of an entire class of animals.

Given these conditions, why not address cases of bird mortality that could be resolved? With all of the knowledge, information, and means to act, why are there only occasionally consistent national or global actions and functional laws for the protection and survival of birds? At the very least, the ever-increasing bird losses via a handful of causes (e.g., cats, windows, power lines) could be addressed. The fate of birds is changeable by national actions, some of which may simply involve passing a new law, amending the penalties and processes for existing laws, or changing the means and processes of enforcing such laws. In small pockets of the world, some effective measures have been shown to work, and these are worth emulating. The decline of birds is a symptom of the changing environment. To minimise the importance of birds is to be blind to the entire concatenation of symptoms of a world in trouble.

In summary, the current state of the world and the often pessimistic prognoses for the future [309,310,311] were probably avoidable had 19th century and mid-20th century published warnings been taken more seriously, e.g., the greenhouse effect was discovered by Eunice Foote in 1856 and, independently, by John Tyndall in 1859, finding that carbon dioxide absorbs and holds heat [414], and as early as 1896 a Swedish scientist (Svante Arrhenius) showed that all fossil fuels (oil, gas, and coal) release carbon dioxide and warm the planet [415]. Instead, vested interests and “climate change deniers” have held back meaningful change. Today, climate change is acute and measurable, and it is no longer at a level of theory but observable fact. The physical changes are obvious—with icecaps melting, glaciers disappearing, and sea levels and temperatures rising, there is plenty of visible evidence of the effects of human impact on the planet as a whole [416,417,418].

In one sense, the claims in the book “*Exposing The Great Climate Change Lie*” by Lynne Balzer [419] ring true. She wrote in her preface that “…solar and wind farms, biofuels and battery-powered vehicles, are not only impractical and unsustainable but actually destructive to the environment”. Yet there is an error in her logic and, while persuasive, the author is hitting out in the wrong direction. Indeed, current designs and methods to utilise renewable sources for energy generation are harmful. For instance, the current wind turbine models, as discussed earlier, are responsible for millions of avian deaths, but the initial idea for the need to find alternative energy sources is sound, and their implementation is urgent. Changes in design are possible, and so is the option (in some cases) of retrofitting devices to limit damage. Design changes are well within current skill levels and have been demonstrated in Spanish inventions of alternative wind turbine design, as also discussed earlier. With the right attitude, change is possible, with a focus on how to generate alternative energy without inflicting damage on an already stressed nature.

The reason why so many new inventions are unsustainable is that too many innovations and products are entirely anthropocentric and have nothing whatsoever to do with solving the problems of climate change or limiting damage to the environment. Some new gadgets and technological inventions are seriously lacking in forethought of what a particular design or product might do to the living world and nature generally, as discussed above. Flying objects such as drones, flying electric cars, and planes are already on the market or have been designed, tested, and readied for mass distribution [380,381,382,383]. Indeed, this human conquering of the airspace should now be included in discussions on habitat loss, because it will predictably do great damage to bird numbers.

Progress will be made when innovations come up with solutions that do no harm to animals and the environment while still addressing the problem at hand. Secondly, progress will be made when laws are either formulated or strengthened and enforced in such a way that they protect what they claim to protect or create an awareness by new laws that enable more protection for birds [407]. Because this has generally not occurred or has not been effectively monitored or enforced [420,421], all living things are declining, and whole bird communities edge closer to extinction, while habitat destruction continues [305].

Problems in energy infrastructure could be substantially reduced by interdisciplinary collaboration. The results achieved by the APLIC, discussed above as an example of meaningful interdisciplinary collaboration (here, on direct mortality of birds caused by power lines), are a shining example of how beneficial such a collaboration can be [213,214]. Given that hundreds of thousands of kilometres of power lines will have to be built for the new power sources, such as solar, wind, and hydro (see Figure 3 above), one may ask whether the companies now gaining contracts in many different countries even know of the Avian Power Line Interaction Committee (APLIC) [213]. Do they know of the APLIC’s way to significantly curtail bird deaths, or has this innovativeness also stayed local and fallen into oblivion? If contractors are aware of the APLIC’s findings and solutions, might any of them be included in the planning stage of power line design across the world? Or will technicians continue to build power lines in the same traditional manner and, thus, continue to guarantee the predictable deaths of further millions of birds?

Generally, finding workable solutions to safeguard birds’ survival, both in the air and at sea, should be one of the key requirements for any new technology. Technological cleverness is no longer enough. Neil Postman, author of the book *Technopoly*, as mentioned earlier, argued some years ago (1995) that technology does not invite a close examination of its own consequences [379]. He also argued that technology, somehow perceived as superior to human actions and thinking, has created a culture without moral foundation [149,379].

Indeed, it would seem that we need new ethical standards. Paramount for animal and human well-being is that a majority position be reached that all animals (from fishes and other water creatures to insects, amphibians, reptiles, birds, and mammals) living in the natural environment today have a right to live and have that right reinforced by respecting their living spaces and, if need be, securing them with human help [72,73,74,417], and that all living things (here, birds) have their relative importance in the web of life acknowledged [6,8,422].

## Figures and Tables

**Figure 1 animals-15-00073-f001:**
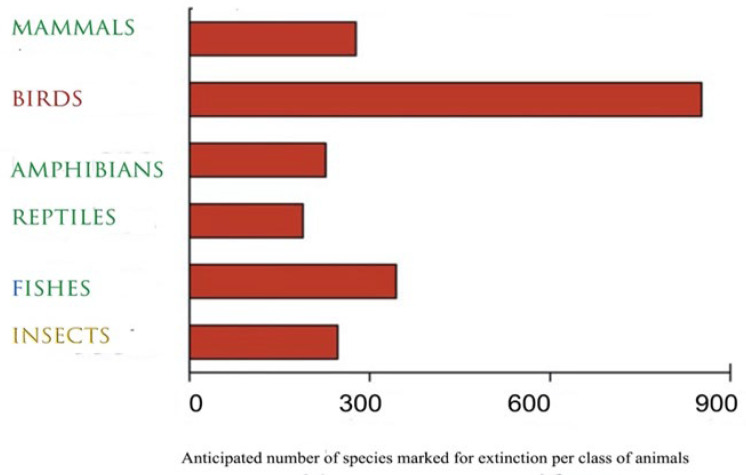
Prognosis of further decline/extinctions concerning near-threatened species (Finn et al. 2023) [14], who based their calculaations on IUCN Red List estimates.

**Figure 2 animals-15-00073-f002:**
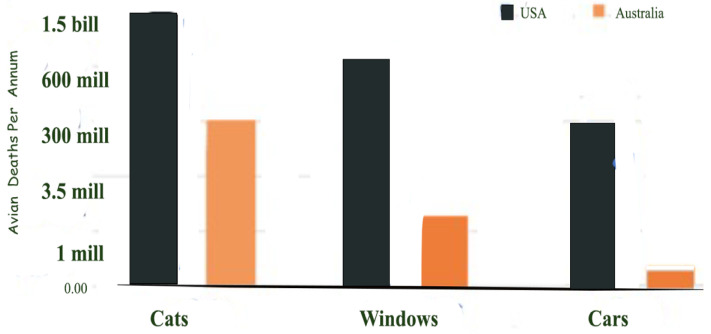
Major causes of annual avian deaths in the USA (black) and in Australia (orange). The USA has more than ten times the population of Australia but a similar landmass (the USA is 1.28 time larger than Australia). In the US, the unimaginable figure of an estimated 1.5 billion birds die per annum as a result of cat attacks. In Australia, the figure is much smaller, which suggests that the scaling down of numbers has to do with the size and density of the human population (see also Table 2 below) [113,114,115].

**Figure 3 animals-15-00073-f003:**
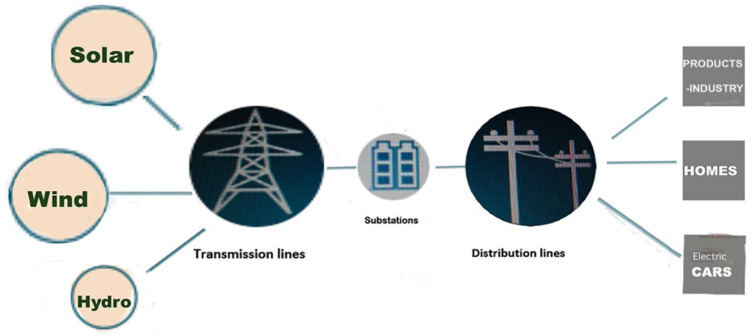
The new powerline network, from electricity generation to consumers, requiring new transmission lines according to the source of power and distribution lines to industry, to homes, and now also to electric cars; adapted from Bernardino, 2021 [202].

**Table 1 animals-15-00073-t001:** Attitudes to where native animals should live, in% of respondents, excerpted from Davies and Webber 2004 [60].

Environment	Definitely Encouraged (%)	Encouraged to Some Extent (%)	Not Encouraged at All %
Suburban backyards	14	50	36
Local parks	23	52	25
Unspoilt bushland	87	12	1

## Data Availability

Data are contained within the article.

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
