# Peer review of "Human-Caused High Direct Mortality in Birds: Unsustainable Trends and Ameliorative Actions"

_animals, 2024, doi:10.3390/ani15010073_

Round 1
Reviewer 1 Report
Comments and Suggestions for Authors
I found this manuscript somewhat provocative. However, it lacks the structure of a typical paper, including sections such as introduction, objectives, methods, results, and conclusions. Consequently, if published, it would be more suitable for use as a commentary type of manuscript.
It's important to consider a separate section on multi and transdisciplinary work. This will allow you to draw more comprehensive conclusions.
In general, I found it excessively lengthy, particularly the conclusion remarks. I think the conclusion must be no more than three paragraphs.

Author Response
Thank you very much for your comments. The author has revised the manuscript. Please see the author's response in the attachment.

Reviewer 2 Report
Comments and Suggestions for Authors
Please see attached comments.

As mentioned in my review, the paper has a conversational, provocative tone that is not appropriate for a scholarly scientific discussion.
Author Response

(The authors gave the same response as above.)

Reviewer 3 Report
Comments and Suggestions for Authors
This is a thorough exploration of an important subject, which I believe should be published in some form but would benefit from several substantial changes to its current form. I therefore recommend the author undertake a major revision to make it suitable for publication in this journal. The author is clearly passionate about addressing threats to bird conservation, a passion I share entirely, and thus I agree completely with her goal of advancing bird protections. However, the paper as currently written struck me as something much more appropriate for a humanities or liberal arts journal than a science journal. It is also apparent that the author is not a native English speaker and the text would benefit from careful editing by a native English speaker skilled in this area.
To make the article suitable for a science journal, several issues need to be addressed, in my view. The author might consider if she prefers to tailor it for an alternative journal (in the social sciences or humanities?) If it is to be published in this journal, which is a science journal, I believe the manuscript needs to be tailored and focused around a clear central question or hypothesis that is addressed. There is a question in the title, of course, but I did not feel this received a structured examination in the paper itself that was satisfactory. This paper provides helpful advice about shortening and focusing manuscripts: https://onlinelibrary.wiley.com/doi/10.1002/ece3.11543
As this paper is a review in a subject with which I am familiar, I found myself searching for what is new and different in the author’s approach, or what the reader will learn. I believe the paper would benefit from a systematic approach to bringing forth key lessons from past research, such as the approach taken in this paper: https://www.mdpi.com/2673-6004/4/2/21
Note that in the above-cited paper, the authors described how they identified the “10 principles” that are the subject of the paper, and also that the paper is categorized as a “perspective” rather than a systematic review. That is, the paper is influenced by the authors’ opinions as well as the evidence from the literature. That strikes me as the case in the present paper as well. Again, I fully agree with the perspectives the author presents, but found myself searching for what is novel in her presentation. The paper thus struck me more as the term paper of a very diligent student than a scientific paper, because it is summarizing past material rather than presenting new findings or evidence.
The paper also makes a great many claims that can be contradicted, depending on the context. For example, that “making laws is cheap.” This can indeed be the case in some contexts, but in others, enormous amounts of money are spent on campaigns to pass laws, so making laws is not necessarily "cheap." Moreover, even if passing a law is relatively simple, enforcing laws is a different matter, as the author notes elsewhere in the paper.
Many laws regarding cats are simply not enforced by governments, because it is too expensive. The same is true with protected areas in tropical Africa, where illegal behavior tends to be extremely common. If the laws are not enforced and people are not aware of them, such as is the case in many countries where I have worked in sub-Saharan Africa, they do not provide conservation benefits and are not sufficient by themselves as a solution.
The simple summary and abstract include claims that may be common in the humanities or journalism (such as references to eating or living the world “to death”) but are less suited to a science paper that is primarily concerned with advancing the state of scientific evidence. The author’s findings and recommendations should be concisely included in the abstract, so that the reader has an idea of the contents of the paper. Moreover, the abstract should contain the key findings of the paper through the analysis and interpretation made by the author, but as it stands currently the abstract simply tells the reader what the author will discuss in the paper, which I find insufficient. Most readers, lacking time, will look to the abstract for a concise summary of results, which they will not find here, and this absence will dissuade many from undertaking to read the paper in its entirety.
As the findings and recommendations are in short supply in the abstract, we must read almost to the end of the paper to find the following statement: “As we know, the effort of those many ornithologists, explaining the ecosystem value of birds for humans for decades now and into the present have barely made a dint in improving the lives of birds. Those who advocate saving species and give their time and energy to such pursuits know that they are fighting a losing battle…” She then quotes another author, who writes, “Very little has changed in the legal landscape the last few decades… despite a 38-fold increase in domestic environmental laws and regulations… the inability to fully implement and enforce these laws remains a significant problem.”
Now the reader may be confused. Is the author saying laws or important, or not? Is she saying there is no point in the efforts of people to save birds, as is implied in her writing above? She continues, saying, “This paper has therefore advocated a small piecemeal approach, addressing documented extreme annual losses of birds and making these known to lawmakers, politicians and relevant industrial groups including issue specific professionals.” Now the reader may be even more confused… aren’t these things already happening? Aren’t such actions the same as those the she has just written are ineffective? And how is working with politicians and industries “piecemeal”? And how would this be expected to succeed when other efforts, according to her previous statements, have failed? The reader is left wondering what the author is really trying to say, on the one hand, and on the other hand, what is new in what she is saying?
The author uses “the royal we” throughout the paper but who is "we" ? The entire population of the world? Scientists? Concerned citizens? There are many countries in the world, such as Australia, Sweden, and the USA, where substantial legislation exists to protect birds and many places where protecting birds is also a cultural tradition; there are also many countries and regions, such as in tropical Africa and Asia, where fast-growing human populations do not feel that protecting wildlife is "their problem"... defining the "we" is important for advancing the positive outcomes the author describes.
For example, one sentence reads as follows: “We ‘only’ need a law, preferably worldwide, that confines cats to the owner’s property (cat runs) and billions of birds could survive.” While I share entirely the author’s concerns about the terrible toll domestic cat predation takes on birds, this sentence is problematic for several reasons. First, the concept of a single “worldwide” law implies that all of the world’s people and nations are governed by a single government, which is contrary to reality. Second, even within a single nation, such as the USA for example, laws governing domestic animals tend to be made at the municipal (city) level, rather than the state or federal level. Third, municipal laws requiring domestic animals to be confined to their owners’ property tend not to be enforced in the case of cats, because such law enforcement is expensive and not a priority for most governments compared to dogs, which can present a public health risk to people, or cattle, which may present traffic hazards, and so on. Thus, the suggestion the author makes does not appear to be one likely to advance the goals she supports. This is only one example, but demonstrates one of many such issues that should be addressed to improve the paper. The author might choose one set issues on which to focus in more depth and thereby split the current manuscript into several papers. Questions about cat predation, for example, certainly merit their own paper and bird conservation might benefit from a manuscript focused on this theme, among others. I have made a few other comments and notes in the attached version of the manuscript.
I believe the author’s effort to understand and summarize all of these threats, questions, and potential mitigation solutions are commendable. However, currently the paper lacks focus and direction, contains many problematic and possibly contradictory statements, and suffers from an overly broad focus that forces a lack of depth in terms of understanding problems and solutions. I therefore again suggest referring especially to the two references provided above to focus and shorten the paper and provide some concise take-aways for readers to advance the understanding of this subject, and ultimately to advance conservation.

It is apparent that the author is not a native English speaker and the text would greatly benefit from careful editing by a native English speaker skilled in this area.
Author Response

(The authors gave the same response as above.)

Round 2
Reviewer 1 Report
Comments and Suggestions for Authors
The typescript lacks a proper article structure, as it only includes an introduction, objectives, methods, results, and conclusions. Therefore, it should be considered a Commentary rather than a full article if it is to be published. I would support its publication as a commentary, but I do not approve of it being published as an article.
On the other hand, I am grateful that the authors have reduced the manuscripts and corrected several comments. It's a good essay.
Author Response

(The authors gave the same response as above.)

Reviewer 2 Report
Comments and Suggestions for Authors
General Comments
Coming in at 67 pages in length, this paper is far too long. In many cases, the text could be cut substantially without losing the ideas being presented. It is written more in the style of a book than a research paper and continues to use problematic wording that is provocative rather than objective. In addition, the new structure of the paper does not follow a typical ms format (introduction, methods, results, discussion, conclusions). The editor should review this for journal requirements.
Specific Comments
· Title. What does “The Gulf Between Anthropogenic Practice and Biodiversity Rhetoric” mean?
· Simple Summary. This section has some very long sentences that should be edited.
· Abstract
- The abstract should present some of the important conclusions that are alluded to in the last two sentences. What specifically did the author find regarding which direct mortality could be “reduced substantially and immediately?” What are some of the “cross disciplinary solutions?” I know the abstract has a strict word limit, but some of the preceding text could be shortened or eliminated.
- What is meant by the statement “Human interaction with birds has never been more positive?” It is in direct contradiction with the rest of the sentence that reads “direct mortality of birds from anthropogenic causes has increased and has led to significant annual losses of birds.” Change to something like “Human appreciation of birds and concern for impacts on birds is very high . . .”
- Change “birds’ ‘ecosystem services’” to “ecosystem services provided by birds.”
· Introduction
- Line 61. Insert “human” between “expanding” and “populations.”
- Line 97-98. Remove the parentheses.
· Section I. The Human-Birds Context.
- The material in this section, which replaces the original Section 2, would be more appropriately presented as a single paragraph in the Introduction of the paper with references to exiting literature.
- If Figure 1 is retained (which I do not recommend), the following changes should be made:
§ The x-axis should be labelled, and the caption should specifically describe what the figure represents e.g., “number of species listed in the IUCN Red List as near threatened.”
§ The text in the caption starting with “Finn et al.” should be moved to the text.
§ I believe caveats should be added to the text describing the results presented in Figure 1 (not the figure caption) explaining that “all taxonomic groups do not receive the same level of evaluation by researchers, and, thus, are not as well-known as birds" and that “the Red List should not be interpreted as a full and complete assessment of the world's biodiversity.”
· Section II. Anthropogenic and Related Causes of Direct Mortality (replaces original Section 3 and 4)
- New text has been added to the Introduction identifying direct mortality as the primary focus of the paper, but the effect that habitat loss (e.g., clearing lands being used by birds) would have on direct mortality of birds was not mentioned as recommended in my comment on the previous version (Section 3.1, line 306). When an area is cleared, birds in that area die during that clearing or later if they cannot find adequate replacement habitat elsewhere. This should be mentioned as a form of direct mortality that is not considered in this paper.
- Concerns remain about wording. A thorough editing is needed. Examples:
§ Line 615, “The news is not so positive for birds.” This is odd as the starting section of a section. News about what?
§ Line 620, “Somehow ensconcing this alternative energy source in the landscape may have falsely led some to think that the topic of climate change implicitly involves all living things and will save the earth.” I’ve read this several times and don’t know what it means.
§ Lines 639-660. The discussion about the motivation behind selection of bladed wind turbines rather than non-bladed designs that are more bird friendly, seems unnecessary. Is it not enough to say that this has occurred without a compelling reason to do so?
§ Lines 700-730. The acronym for “concentrating solar power” is defined 6 times on the same page.
- A section on solar energy impacts has been added. It contains a description of oil field impacts that produce 500,000 to 1 million annual fatalities begging the question of why this is not included as a major cause of direct bird mortalities in this paper. This section also discusses other ecological impacts of solar energy production that are outside the scope of this paper.
- A section on geothermal energy impacts has been added, but the title of the section is “Geothermal and hydropower.” There is no mention of hydropower impacts in the section, but there is a discussion of fossil fuel impacts (not related in any way to geothermal) and a mention of nuclear energy as a renewable energy type, which it is not.
· Section III. Direct Consequences of Deaths and Possible Mortality Reductions (replaces original Sections 6-8)
- This section should be reduced considerably in length. While it is interesting to read, it is more suitable for a book than a research paper. The ideas could be stated more succinctly with references to supporting literature.
- Line 1532. The question asked here “why have birds not captured the imagination?” is incorrect. No group up of organisms has captured people’s imagination than birds as witnessed by many bird-centered activities such as bird feeding, bird watching, bird tourism, and citizen science programs throughout the world.
Author Response

(The authors gave the same response as above.)
